# In-situ electro-responsive through-space coupling enabling foldamers as volatile memory elements

Jinshi Li[1], Pingchuan Shen[1], Zeyan Zhuang[1], Junqi Wu[1], Ben Zhong Tang [2] & Zujin Zhao [1]✉

Voltage-gated processing units are fundamental components for non-von Neumann architectures like memristor and electric synapses, on which nanoscale molecular electronics have possessed great potentials. Here, tailored foldamers with furan–benzene stacking (f-Fu) and thiophene–benzene stacking (f-Th) are designed to decipher electro-responsive through-space interaction, which achieve volatile memory behaviors via quantum interference switching in single-molecule junctions. f-Fu exhibits volatile turn-on feature while f-Th performs stochastic turn-off feature with low voltages as 0.2 V. The weakened orbital through-space mixing induced by electro-polarization dominates stacking malposition and quantum interference switching. f-Fu possesses higher switching probability and faster responsive time, while f-Th suffers incomplete switching and longer responsive time. High switching ratios of up to 91 for f-Fu is realized by electrochemical gating. These findings provide evidence and interpretation of the electro-responsiveness of non-covalent interaction at single-molecule level and offer design strategies of molecular non-von Neumann architectures like true random number generator.

Exponential proliferation of data arouses the urge of breakthroughs beyond the von Neumann bottleneck on nanoscale electronics to satisfy high integration density, low energy consumption, and rapid possessing speed. These days, non-von Neumann architectures exert thriving potential on signal processing, machine learning, deep learning, and stochastic computing[1–3]. On one hand, in-memory computing[4–6] is an effective alternative to avoid the shuttling limitation between the processing and memory units and thus speed up responsiveness and lower energy dissipation. In this way, memristors[7–9] are at the core of in-memory computing to perform computational tasks within the memory itself, which exhibit voltage-triggered inherent resistance variations and remembrance of current flows' history. On the other hand, neuromorphic computing[10–12] is another solution to deal with complicated and massive information

emulating neuro behaviors with high transport efficiency and low power consumption. In neurons, voltage-gated channels achieve intracellular action potential to transport stimulus along axon while electric synapses enable intercellular communication through gap junctions, providing references to construct neuro circuit[13,14]. Intriguingly, both in-memory and neuromorphic computing come to a common need of electric elements in possession of voltage-triggered variations, encouraging further development of electronics with various mechanisms and properties to satisfy diverse applications.

Such voltage dependent memristive and biomimetic devices are currently performed in exploitation of energy-inefficient silicon-based components, or nanocells consisting of phase-change[15], metal-oxide[16], magnetic[17], or ferroelectric[18] materials. In that case, molecular electronics grow as appealing candidates owing to their unique advantages

[1]State Key Laboratory of Luminescent Materials and Devices, Guangdong Provincial Key Laboratory of Luminescence from Molecular Aggregates, South China University of Technology, Guangzhou 510640, China. [2]School of Science and Engineering, Shenzhen Institute of Aggregate Science and Technology, The Chinese University of Hong Kong, Shenzhen, Guangdong 518172, China. ✉e-mail: mszjzhao@scut.edu.cn

including footprint minimization, low cost and multiple responses under electric field[19–21]. Different stable conducting states under varied electric fields are the basis of voltage-triggered transformation. Electric switching for molecule usually relies on electro-induced isomerism[22,23], electro-orientation[24,25], or electrochemical redox[26,27], which aggressively change the structural or charged states in need of high turn-on voltage. Alternatively, voltage-induced quantum interference (QI) switching[28–30] is capable of achieving large switching ratio and low subthreshold swing, using the unique quantum effects on mesoscopic level to acquire distinct conductance change between constructive and destructive QI states within a small voltage range. But current regulation is mainly dependent on electrochemical gating by tuning the relative positions between molecular orbitals and the Femi level, which usually require additional gating electrodes to realize efficient modification as transistors but not two-electrode memristor.

In this work, we report a novel type of voltage-triggered QI switching depending on through-space interaction and its responsiveness dominated by orbital polarization under electric field in heterocycle-benzene stacking foldamer systems, revealing their potential application in true random number generator (TRNG)[31] and neuron-mimicking signal transmission[13] (Fig. 1a). In consideration of dipole moments and structural symmetry, *f*-Fu and *f*-Th are designed and characterized as foldamers that antiparallelly fold into furan–benzene stacking and thiophene–benzene stacking, respectively. During scanning tunneling microscopy break junction (STM-BJ)[32,33] measurement, *f*-Fu exhibits relatively lower conductance while *f*-Th shows comparatively higher charge transport at 0.1 V. Intriguingly, under intensified electric field, *f*-Fu displays turn-on feature with an additional high conductance (HC) state, while *f*-Th performs counterintuitive turn-off feature with an extra low conductance (LC) state. The electro-polarized through-space orbital interaction under intensified

electric field dominates stacking malposition during structural relaxation, responsible for the switching of through-space coupling that determines the transformation of QI and molecular conductance. The switching of *f*-Fu and *f*-Th is sensitive to circumstance with different responsive time constants depending on sub-angstrom deformation. *f*-Fu realizes complete transformation during the hovering of molecular junctions based on two different mechanisms, possessing higher switching probability and faster responsive time, while *f*-Th suffers incomplete switching and slower responsive time owing to increased dispersion and decreased electrostatic repulsion. High switching ratios of up to 91 for *f*-Fu is achieved by the combination of electro-polarization and electrochemical gating. These findings provide evidence and interpretation of the electro-responsiveness of non-covalent interaction at single-molecule level and offer design strategies of TRNG to encrypted communication and biomimetic action potential to transport undecayed signals based on these foldamers.

## Results

### Electro-responsive in-situ conductance switching

Previous studies of external electric field on stereochemical selectivity and catalysis, protein denaturation and DNA recognition enlighten that electric response depends on the relative size and orientation of the dipole moment and polarizability terms in energy expression[34–36]. In this regard, the symmetric benzene–benzene stacking is not a suitable model to discuss electro-responsiveness because of its negligible dipole moment (Fig. 1b). In that case, incorporating heterocycles with proper dipole moments in stacking model is a practicable solution. The preliminary analysis on electrostatic potential distribution[37] also supports that unbalanced charge distribution in heterocycle–benzene stacking with larger dipole moments (Fig. 1b), indicative of higher

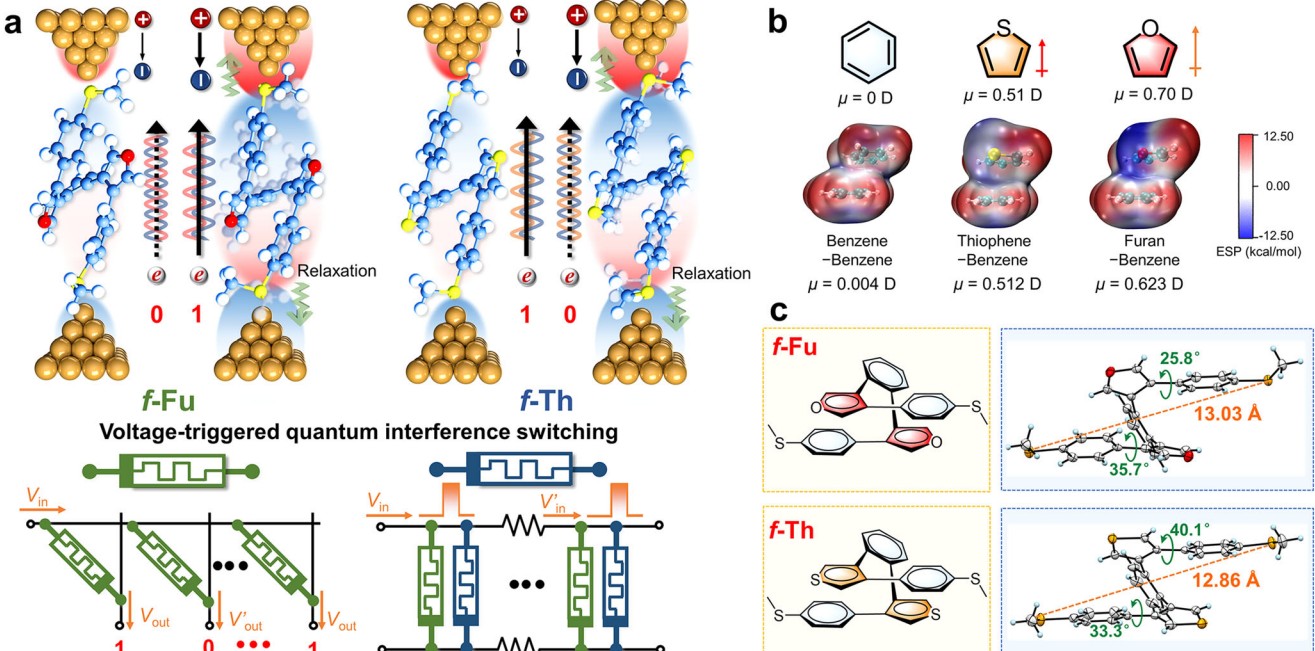

**Fig. 1 | Working mechanism and design principle for electro-responsive quantum interference switching foldamers in this work. a** Schematic illustration of polarization induced electro-responsive quantum interference switching in heterocycle-benzene stacking foldamers and conceptualized demonstration of their possible application on memory elements as TRNG and axon-like voltage-gated channels. The number 0 labels the low conductance state while the number 1 labels the high conductance state. The green memristor represents *f*-Fu while the blue one represents *f*-Th. **b** Chemical structures of benzene, thiophene and furan, and the electrostatic potential distribution (ESP) of benzene-benzene stacking and heterocycle-benzene stacking models with their dipole moments. **c** Chemical structures of *f*-Fu and *f*-Th and their corresponding single crystal structures with labeled torsion angles of the stacked arms and labeled distance between sulfur atoms of thiomethyl groups.

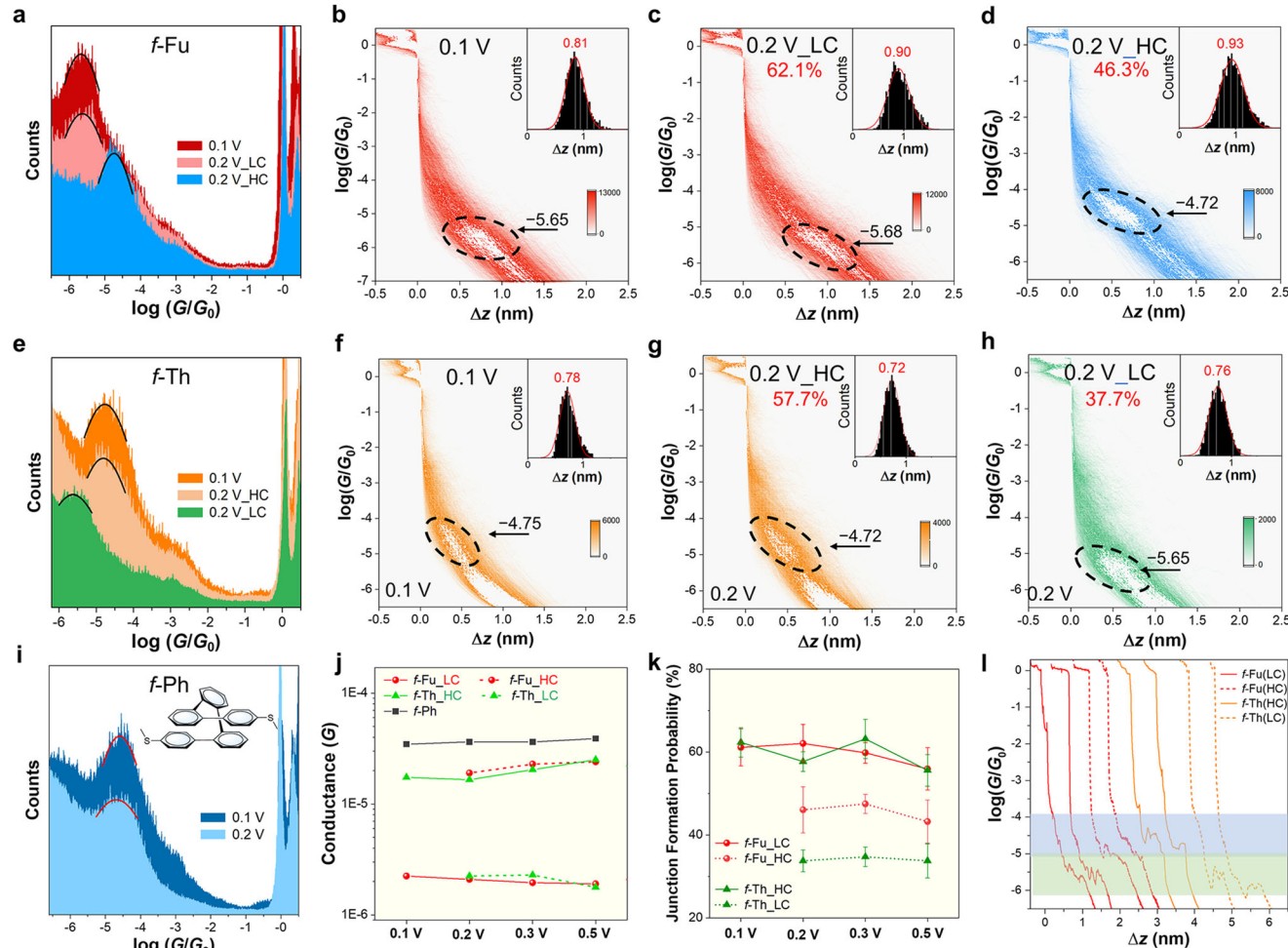

**Fig. 2 | Break junction conductance measurement of foldamers in tetra-hydrofuran/mesitylene (THF/TMB, 1:4, *v/v*) mixture. a** 1D conductance histograms of *f*-Fu at 0.1 V and 0.2 V. 2D conductance–displacement histograms of *f*-Fu at 0.1 V (**b**) and 0.2 V (**c**, **d**) with inserted relative displacement distribution histogram. The black dashed lines circle out the density clouds of conductance distribution with labeled conductance peaks. Junction formation probabilities for HC and LC states is also provided. **e** 1D conductance histograms of *f*-Th at 0.1 V and 0.2 V. 2D conductance–displacement histograms of *f*-Th at 0.1 V (**f**) and 0.2 V (**g**, **h**) with inserted relative displacement distribution histogram. Junction formation probabilities for HC and LC states is also provided. **i** 1D conductance histograms of *f*-Ph at 0.1 V and 0.2 V. **j** Statistic conductance data of *f*-Fu, *f*-Th and *f*-Ph under different applied biases. **k** Statistic junction formation probabilities of *f*-Fu and *f*-Th under different applied biases. The error bars are the standard deviation of multiple results for junction formation probabilities in conductance measurment over three times. **l** Representative conductance–displacement traces of different states of *f*-Fu and *f*-Th at 0.2 V. The blue shaded region represents high conductance states while the green one represents low conductance states.

sensitivity to electric field. Given that effective through-space interaction asks stacking units for matching energy level alignment, dual heterocycle–benzene stacking is designed as upside-down linkage to ensure molecular symmetry in foldamers.

*f*-Fu and *f*-Th with furan–benzene and thiophene–benzene stacking conformations, respectively, are synthesized and characterized (Supplementary Note 5), unanimously holding two stacked aromatic arms with an *ortho*-substituted benzene ring (Fig. 1c, Supplementary Figs. 4, 55 and 56). The single crystal structures confirm that both foldamers adopt anti-parallel conformations, in which the asymmetric stacking with through-space conjugation[38,39] is successfully formed. *f*-Fu and *f*-Th clearly exhibit singular molecular conformation in solutions, as evidenced by the absence of proton signals from other conformations (Supplementary Fig. 6). In comparison to the regular aromatic proton's signals at *ca.* 7–8 ppm, the proton signals of $H_a$ and $H_b$ for these foldamers are apparently upfield-shifted to *ca.* 6.0–6.5 ppm, due to the aromatic shielding effect of the stacking[40–42]. But in general, the proton signals of *f*-Th and *f*-Fu are downfield-shifted relative to those of control foldamer *f*-Ph with benzene–benzene stacking because of weakened shielding effect due to the smaller ring

currents of thiophene and furan[43]. This ¹H NMR information manifests *f*-Th and *f*-Fu possess stable heterocycle–benzene stacking in solution state as well.

Molecular conductance measurement is performed by STM-BJ technique and the purity of all the studied molecules in this work is confirmed by high-performance liquid chromatography (Supplementary Fig. 7). *f*-Fu has sole inefficient conductance of $10^{-5.65\pm0.04}\,G_0$ at 0.1 V (Figs. 2a and 2b). However, two conductance peaks referring to different states are observed for *f*-Fu in the measurement at 0.2 V (Fig. 2a, c, d). For *f*-Fu at 0.2 V, there is a major LC peak of $10^{-5.68\pm0.02}\,G_0$, consistent with its conductance at 0.1 V, but an additional minor HC peak of $10^{-4.72\pm0.03}\,G_0$ is recorded. The conducting performance for *f*-Th is the opposite, where *f*-Th has single higher conductance of $10^{-4.75\pm0.03}\,G_0$ at 0.1 V (Fig. 2e, f) comparing to *f*-Fu. At 0.2 V, for *f*-Th, the HC state of $10^{-4.74\pm0.01}\,G_0$ stays as the majority and the extra LC peak of $10^{-5.65\pm0.03}\,G_0$ is numerically akin to that of *f*-Fu, indicative of similar conducting behavior (Fig. 2g, h). The two-dimensional (2D) histograms reveal that all the emerging states own tilted conductance density clouds, indicative of unstable junctions corresponding to common weakly coupled through-space charge transport. The observed junction lengths

of *f*-Fu and *f*-Th are 1.31 and 1.28 nm after adding 0.5 nm snap-back distance[44,45] for calibration (Fig. 2b, f) at 0.1 V, consistent with the distances between two sulfur atoms (S–S distances) of two thiomethyl (SMe) anchors measured in their crystals (Fig. 1b). The junction length of LC state of *f*-Fu is 1.40 nm, while that of HC state becomes slightly longer (1.43 nm) at 0.2 V. For *f*-Th, the junction lengths of HC and LC states are 1.22 and 1.26 nm, respectively. That is to say, the additional conducting states are ascribed to the longer contacting lengths that are induced by intensified electric field. The possibility of impurity-induced extra conducting states is ruled out at first by the characterization of high-performance liquid chromatography (Supplementary Fig. 7) to ensure high purity of the tested molecules. Additionally, according to the ex-situ electrochemical measurement (Supplementary Fig. 43), a low applied voltage of 0.2 V is incapable of triggering the electrochemical redox reaction of these foldamers, indicating that no impurity is generated during the conductance measurement.

The statistic results reveal similar phenomena under higher biases (Fig. 2j, k). When the applied biases are over 0.2 V, *f*-Fu holds original LC states as main states and additional HC states appear, while *f*-Th maintains initial HC states and minor LC states arise. And, the junctions' lengths of minor states become slightly longer than those of the original states (Supplementary Note 2.1, Supplementary Figs. 8 and 9). Both HC states mildly increase while both LC states slightly decrease under intensified electric filed. As a result, the switching ratios between two states are gradually increased at higher biases and achieve 14 at 0.5 V. The junction formation probabilities of the major states (LC for *f*-Fu and HC for *f*-Th) are about 60% without obvious difference. But the minor LC states of *f*-Fu own higher probabilities around 40% than the minor HC states of *f*-Th around 30%, which suggest the on/off ratios are about 66% and 50% for *f*-Fu and *f*-Th, respectively, implying the transformation from major states to minor states is easier for *f*-Fu to achieve. Also, the junction formation probabilities of all states decrease at 0.5 V because the intensified electric field would aggravate Brownian movement with more drastic thermal fluctuation and thus disturb the stability of molecular junctions. In contrast, the control molecule (*f*-Ph) with benzene–benzene stacking[46] remains nearly the same conductance in all measurements (Fig. 2i and Supplementary Fig. 10). The co-existence of conducting states is still realizable under intensified negative biases but with lower junction formation probabilities for both conducting states (Supplementary Note 2.2, Supplementary Figs. 11, 12 and 13), and an extra conductance stemming from the coupling of heterocycle and electrodes can be found in the experiment under intensified negative biases, which is further discussed in Supplementary Information.

In STM-BJ measurement, the tip-lifting rate is positively controlled by the piezo rate. Herein, a piezo rate of 30 nm s⁻¹ is adopted in all experiments after optimization (Supplementary Note 2.3, Supplementary Figs. 14, 15, and 16) to ensure that *f*-Fu, *f*-Th and *f*-Ph all show distinct single conductance peak at 0.1 V and reduce misleading effect from *f*-Ph's minor conformer and tips' stretching strength. The feature of switching under higher biases can still be observed in the measurement with higher piezo rates (Supplementary Note 2.3, Supplementary Fig. 17), indicating the additional conducting states are more likely to be induced by voltage rather than mechanical force. The conductance–displacement traces (Fig. 2l) confirm that both HC and LC states for *f*-Fu and *f*-Th appear individually rather than simultaneously during STM-BJ processes, suggesting that the minor states do not stem from the direct stretching of molecular junction with initial major states by the moving gold tip.

To rule out the possibility of multi-anchors for *f*-Th in particular, the conductance of two control molecules with one SMe anchor (*f*-Th-H) and without SMe anchors (*f*-Th-2H) are performed for comparison (Fig. 3a and Supplementary Fig. 5). *f*-Th-H exhibits two blurry conductance peaks around $10^{-3}$ and $10^{-5}$ $G_0$, probably due to the anchoring of SMe and thiophene to the electrodes and through-space charge

injection via terminal benzene, respectively. The details about this are provided in Supplementary Information (Supplementary Note 2.4, Fig. 3b, and Supplementary Fig. 18). *f*-Th-2H adopts a co-parallelly folded conformation but no distinct conductance peak is detected, suggesting two thiophenes in foldamers are difficult to anchor gold electrodes simultaneously (Fig. 3c). Their conductance results are found to be totally different from that of *f*-Th, excluding the possibility of the connection between gold electrode and thiophene in *f*-Th, probably due to that the stacking geometry shields the sulfur atoms of thiophenes and thus the exposed SMe anchors are more likely to attach to electrodes. For *f*-Fu, the larger electronegativity of oxygen atoms is not conducive to attaching to gold electrodes. These findings confirm that the obtained conductance results of *f*-Th and *f*-Fu stem from the junctions formed by attaching SMe anchors to electrodes.

To evaluate the conductance based on sole through-bond pathway, *Z*-shaped unstacked isomers (*Z*-Fu and *Z*-Th) are prepared (Fig. 3d and Supplementary Fig. 5). Among them, *Z*-Th exhibits the characteristic of multiple conductance resulting from the disturbance of thiophene-anchored junctions because of the lack of shielding effect in unstacked structure (Supplementary Note 2.4, Supplementary Figs. 19 and 20). But here, we focus on the conductance originating from the contacting configuration with two anchored SMe groups. Actually, *Z*-Fu and *Z*-Th are found to have quite similar conductance of *ca.* $10^{-5}$ $G_0$, lower than that of *f*-Th but higher than that of *f*-Fu (Fig. 3e, f). Furthermore, these two molecules do not perform turn-on or turn-off features at higher applied biases and all conducting states remain nearly the same at different biases (Supplementary Figs. 21–24), declaring the conductance differences between *f*-Fu and *f*-Th and the conductance switching with additional states do not result from the differences of through-bond pathways. Overall, the co-existence of two completely different conducting states under intensified electric field is a prominent feature for *f*-Fu and *f*-Th with asymmetric heterocycle−benzene stacking. Besides, the influence of intermolecular interaction is discussed in the Supplementary Information. The foldamers are inclined to adopt twisted conformation to ensure sufficient intramolecular through-space stacking so that the intermolecular through-space stacking is difficult to form because of the steric hindrance from the twisted conformations (Supplementary Note 2.5, Supplementary Figs. 25−29).

Theoretical simulation (Supplementary Note 3) is adopted to unveil the nature of different conducting behaviors for these foldamers. Structural relaxation is observed under electric field with elongated S−S distances (Supplementary Fig. 30), which may be the key to explain the occurrence of multiple conducting states. Foldamers stabilize their conformations via intramolecular through-space interaction. Based on the energy differences between the unfolded and folded conformations (Supplementary Fig. 31), the folded conformation of *f*-Fu has the largest decrement in energy, suggesting the best stability and strongest intramolecular non-covalent interaction. *f*-Th comes next with medium stability while *f*-Ph has the smallest energy difference, indicative of least stability corresponding to multiple conformers in NMR results. Moreover, the overall intramolecular non-covalent interaction can be divided into two types (Fig. 4a). One is the π-π offset stacking within two arms (Type I) and the other is edge-to-face interaction among the central benzene and the terminal benzene rings (Type II). To demonstrate the mechanism of these through-space interactions, the ensemble interaction energy is decomposed[47] as Eq. (1) using the scheme proposed by Morokuma[48], Ziegler, and Rauk[49]:

$$E_{int} = E_{elec} + E_{Pauli} + E_{disp} + E_{orb} \qquad (1)$$

where $E_{elec}$ corresponds to the direct electrostatic Coulomb interaction, $E_{Pauli}$ represents the Pauli exchange repulsion and is responsible for steric repulsion, and $E_{disp}$ accounts for the dispersion corrections. The associated orbital interaction energy $E_{orb}$, known as the induction

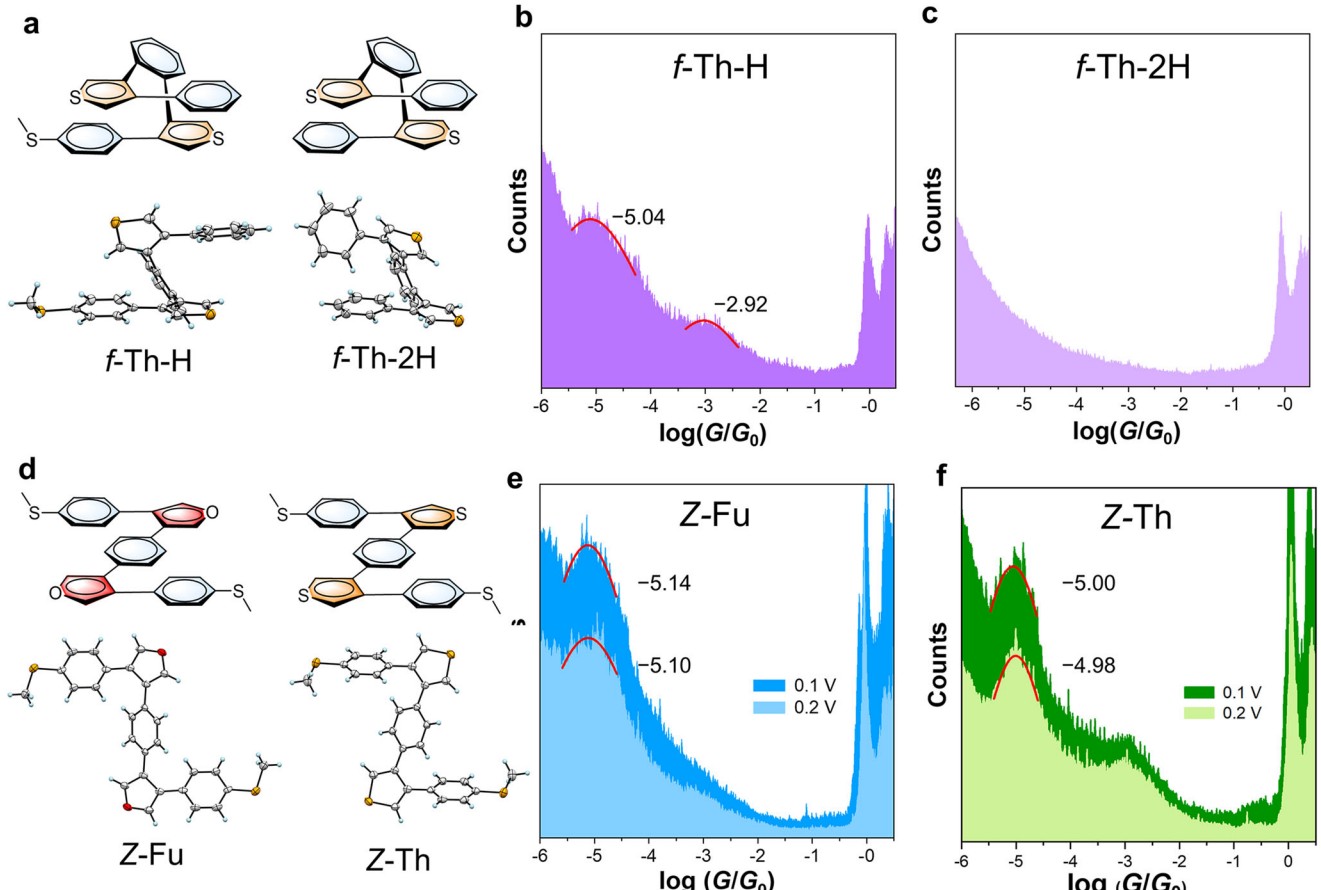

**Fig. 3 | The structures and break junction conductance measurement of control molecules in tetrahydrofuran/mesitylene (THF/TMB, 1:4, $v/v$) mixture.** **a** Chemical structures and crystal structures of $f$-Th-H and $f$-Th-2H. 1D conductance histograms of $f$-Th-H (**b**) and $f$-Th-2H (**c**) at 0.1 V with red fitted conductance peaks and labeled conductance values. (**d**) Chemical structures and crystal structures of $Z$-Fu and $Z$-Th. 1D conductance histograms of $Z$-Fu (**e**) and $Z$-Th (**f**) at 0.1 V and 0.2 V with red fitted conductance peaks and labeled conductance values.

term, accounts for electron pair bonding, empty/occupied orbital mixing on one fragment due to the presence of another. Since principal wavefunction methods do not support the calculation of intramolecular orbital interaction, we adopt the energy decomposition analysis based on molecular force field, which just provides former three terms but not $E_{orb}$ requiring wavefunction analysis. But the interaction of orbitals can be visualized as electron clouds of molecular orbital to proceed evaluation. Herein, Eq. (1) is simplified as

$$E_{int} = E_{elec} + E_{Pauli} + E_{disp} \qquad (2)$$

The binding energies of π–π stacking in optimized $f$-Fu and $f$-Th in gas phase decrease due to the smaller sizes of furan and thiophene with weaker dispersion, and the stronger Pauli repulsion in $f$-Fu further reduces the interacting intensity of furan–benzene stacking. However, the binding energies of edge-to-face interaction increase in $f$-Fu and $f$-Th primarily because of the weakened Pauli repulsion (Fig. 4a and Supplementary Table 1), which is responsible for the enhanced stability in heterocycle–benzene stacking foldamers. The energy decomposition reveals that the dispersion energy possesses dominant contribution to intramolecular through-space interaction.

Based on energy decomposition, the evolvement of through-space interacted energies under electric field is evaluated. To illustrate the energy change more clearly, the direction of uniform electric field is set along the axis of S–S linkage, though the real electric field in STM-BJ measurement is not uniform and difficult to simulate in quantum

chemistry. When molecules do not relax under electric field (0.002 au), the π–π stacking in $f$-Fu destabilizes and the binding energy decreases while that in $f$-Th possess stronger stability. However, both $f$-Fu and $f$-Th re-organize their conformations with longer S–S distances to adjust to the electric field and strengthen binding energies (Fig. 4a, Supplementary Fig. 30 and Table 1), which arouses our curiosity to dig out the inherent nature of electro-responsiveness.

Among three terms in Eq. (2), electrostatic energy is directly affected by electric field and the others are just relative to the atomic positions. The electrostatic potential undergoes re-distribution under electric field since these foldamers are polarized (Fig. 4b and Supplementary Fig. 32). Under intensified electric field, the negative potential will be more concentrated on the end attaching to the positive electrode and the positive potential is on the other end attaching to negative electrode. During polarization, it is observed that the re-distributed negative potential and the negative potential on heteroatoms will mutually repulse, ultimately leading to the malposed stacking and thus structural relaxation. In terms of energy, Coulomb repulsion is observed within the stacked arms with positive-valued electrostatic energy, while Coulomb attraction exists in the edge-to-face interaction with negative-valued electrostatic energy (Fig. 4b and Supplementary Table 1). When molecules do not relax under electric field, the π–π stacking repulsion in $f$-Fu enhances and destabilize the foldamer and the simultaneously enhanced edge-to-face attraction cannot compensate the repulsion, ultimately leading to the displacement of π–π stacking. Therefore, after relaxation, non-covalent binding energy of $f$-Fu increases by reducing coulomb repulsion. Situation

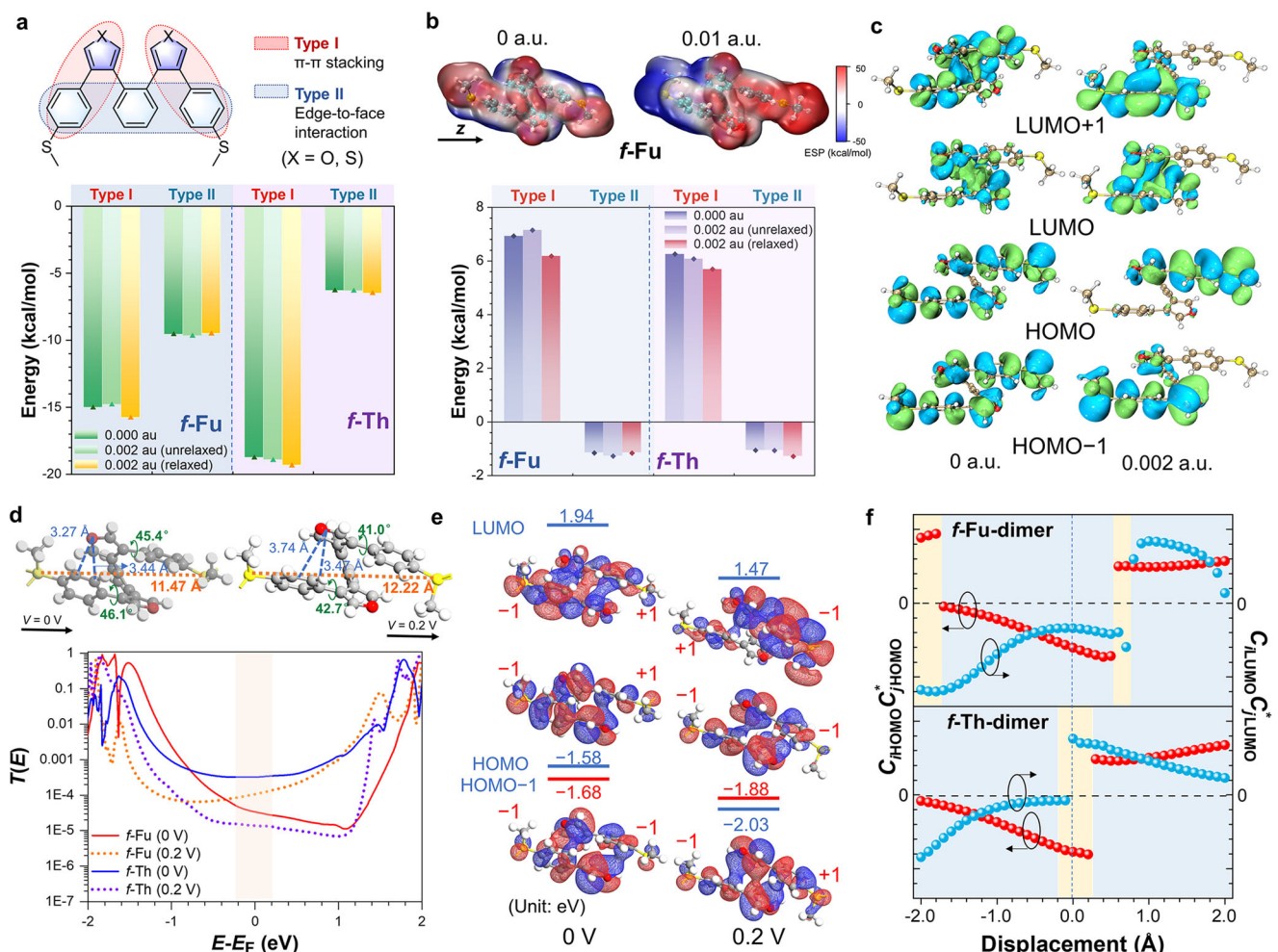

**Fig. 4 | Theoretical simulation based on energy decomposition and orbital analyses. a** Demonstration of two types of non-covalent interaction in foldamers on the upper panel. **b** Electrostatic potential distributions of *f*-Fu under electric field in the intensity of 0 and 0.01 au on the bottom panel. The black arrow refers to the direction of simulated electric field. Energy profile of electrostatic interaction for *f*-Fu and *f*-Th based on gas phase optimized structure under 0 and 0.002 au eletric field and the relaxed structure under 0.002 au eletric field on the bottom panel. **c** Molecular orbitals of *f*-Fu based on density functional theory under different intensities of oriented electric field. **d** Comparison of optimized junctions' structures for *f*-Fu at 0 and 0.2 V on the unpper panel with labeled atoms' distances and

torsion angles. The black arrow refers to the direction of simulated electric field.- Transmission functions of *f*-Fu and *f*-Th at 0 and 0.2 V on the bottom panel. **e** molecular projection self-consistent Hamiltonians and level-crossing of occupied levels for *f*-Fu under different electric fields. The +1 figures correspond to the positive signs of expansion coefficients and the −1 figures correspond to the negative signs. **f** The HOMO and LUMO coupling values of stacking arms with different displacement for *f*-Fu-dimer and *f*-Th-dimer. The blue zones indicate DQI while the yellow zones indicate CQI based on the products' signs of HOMO and LUMO couplings. The zero-point is set as the stacking arms deconstructed from optimized junction configurations of foldamers.

in *f*-Th is different, and the π–π stacking repulsion and edge-to-face attraction are not the driving elements for *f*-Th's relaxation since they are both stabilized under electric field in no need of relaxation.

The last term of orbital interaction is considered to explain structural relaxation based on electron cloud distribution of orbitals as well (Fig. 4c). Without electric field, the orbital distribution is balanced between two stacked arms with bonding and anti-bonding interactions based on hole and electron couplings. These couplings guarantee effective orbital mixing and robust stacking. Once electric field applies, the orbitals are polarized, then the electron clouds re-distribute and usually concentrate on one single arm. The coupling between stacked arms is weakened with less orbital mixing because of the break in original symmetry, subsequently followed by the malposition of stacking. The orbital polarization, found in all foldamers (Fig. 4c and Supplementary Fig. 33), is the primarily diving force for their structural relaxation. To conclude, electric field drives foldamers to relax their folded structures primarily by orbital polarization, but for *f*-Fu, the unstable electrostatic repulsion simultaneously assists relaxation

corresponding to higher formation probabilities of the minor states. Hence it is able to deduce that the extra states for foldamers originate from the newly-appeared relaxed conformations. But the essential of conductance differences between original conformations and relaxed conformations needs in-depth digging in the following discussions.

As for junction simulation, similar structural relaxation under intensified electric field can also be found, which is similar to theoretical analyses in gas phase but displays larger structural deformation. The contacting geometries of optimized junctions display that the torsion angles of the stacked arms decrease and the S–S distances referring to molecular junctions' lengths increase by less than 1 Å (Fig. 4d and Supplementary Fig. 34) under applied biases, which is in good agreement with the elongated lengths of the junctions in experiments. *f*-Fu performs strongest relaxation with the elongation of 0.7 Å, indicative of enhanced sensitivity. The calculated transmissions of optimized molecular junctions' configuration (Fig. 4d) at zero bias indicate that *f*-Fu possesses destructive quantum interference (DQI) with the appearance of antiresonance, while *f*-Th shows constructive

quantum interference (CQI) with symmetrical transmission. By comparing transmission functions of $f$-Fu at 0.2 V to those at the zero-bias, the antiresonance valley referring to DQI disappears with applied biases and the transmission around $E_F$ increases. $f$-Th exhibits totally inverse trend that the antiresonance valley shows up with suppressed conductance under applied bias. In contrast, $f$-Ph remains symmetric transmission as CQI under higher biases (Supplementary Fig. 35). $f$-Fu and $f$-Th possess converse switching mode that $f$-Fu transforms DQI into CQI under high biases while $f$-Th is the opposite. Therein, it is reasonable to deduce a chain of causality that intensified electric field re-organizes non-covalent interaction of foldamers, resulting in structural relaxation to malposed the stacked arms, which subsequently changes QI pattern and brings about the extra conducting states.

The inherent relation between structural relaxation and QI switching can be demonstrated by molecular projection self-consistent Hamiltonians. Ensemble QI can be regarded as the comprehensive result of the QI among molecular orbitals where interferences involving frontier orbitals contribute significantly[50,51], indicating orbital analysis is the key to decipher QI switching induced by structural relaxation. Since the transmission and the transmission pathways[52,] together with MOs' alignment remain almost the same for the dimers in comparison with the foldamers (Supplementary Fig. 35–38), all the foldamer models are simplified as dimer models for convenience. According to Green's function, the opposite signs of the products of the frontier orbitals' coupling at anchoring sites, $C_{iHO}C_{iHO}^*$ and $C_{iLU}C_{iLU}^*$, indicate CQI while their same signs refer to DQI[50,51]. At zero biases' simulation, the identical signs of $C_{iHO}C_{iHO}^*$ and $C_{iLU}C_{iLU}^*$ based on through-space orbital interaction for $f$-Fu suggest DQI and thus transmission cancellation. In the elongated junction of $f$-Fu model at 0.2 V, the asymmetric density distribution on anchored sulfur atoms for HOMO and LUMO is also observed, confirming the existence of electro-polarization. But more importantly, the HOMO and HOMO−1 exchange positions[53] in comparison to the original model (Fig. 4e and Supplementary Fig. 39 and 40), transforming the negative-valued HOMO coupling ($C_{iHO}C_{iHO}^*$) into the positive-valued one and thus switching DQI into CQI. In contrast, deducing from the opposite signs of $C_{iHO}C_{iHO}^*$ and $C_{iLU}C_{iLU}^*$, $f$-Th constructively transports charges owing to CQI at zero bias, so does $f$-Ph (Supplementary Fig. 39). The level-crossing-induced transformation of the value of $C_{iHO}C_{iHO}^*$[53,54] can explain the transformation from CQI at zero bias to DQI at 0.2 V for $f$-Th as well (Supplementary Fig. 40), which does not occur in $f$-Ph. The previous reports[53,55] on alternate through-space QI switching via the malposition of stacking enlighten us to keep digging from the frontier orbitals' coupling with different displacements (Fig. 4f). The dimer models deconstructed from original optimized foldamers (Supplementary Fig. 41) are set as zero-point, and one arm translate forward (turning away from the other) and backward (getting close to the other) along the S–S axis with a step size of 0.1 Å. Both HOMO and LUMO couplings ($C_{iHO}C_{iHO}^*$ and $C_{iLU}C_{iLU}^*$) reverse signs periodically, leading to the change of their products' signs and thus the switching of QI pattern. $f$-Fu experiences QI flipping with forward elongation around 0.6 Å with HOMO coupling reverses while $f$-Th switches QI with forward elongation around 0.3 Å, supporting that structural relaxation induces QI transformation. In this way, the elongated offset stacking is responsible for the conversion of occupied orbitals and the reversion of HOMO coupling values, which brings about the transformation of QI pattern explaining the extra existing states[55,56]. In practical dynamic break junction, however, the orientation of molecules relative to electric field is not as fixed as that in simulation, in which the direction of electric field is set along the straight line connecting two anchoring sulfur atoms. The polarization only works on polaron's vector component in the direction of electric field[34], and the structural reorganization is not always effective enough to ensure the complete QI switching.

In short, the electro-responsiveness of intramolecular through-space interaction is the key to conductance switching under intensified electric filed. The electro-polarization primarily accounts for the re-distribution of non-covalent electrostatic and orbital interaction, inducing sub-angstrom structural relaxation. The staggered stacking arms further results in the transformation of through-space coupling with level-crossing and thus in-situ QI switching with significant conductance difference.

## Environment- and time-dependent switching kinetics

Transmission simulation discloses that the antiresonance valleys for the foldamers lie close to the LUMO (Fig. 4d) but not at the $E_F$ because the electron and hole coupling based on through-space conjugation is unbalanced. Their frontier orbitals' resonances are quite far away from the $E_F$ due to wide band gaps, which are not capable of inducing sufficient suppression of conductance based on DQI and effective enhancement of conductance based on CQI. Therefore, the switching ratios for QI transformation are limited to about one order of magnitude. To fully exploit through-space QI and magnify the switching ratio between HC and LC states, the two-electrode electrochemical gating measurement[57,58] is further applied to shift MOs' relative positions of foldamers to $E_F$. In electrolytic environment full of freely migrated ions, when the gold tip is biased negatively, the antiresonance near LUMO for DQI and resonance of LUMO for CQI move towards bias window (Supplementary Note 4, Fig. 5a and Supplementary Fig. 43). Consequently, the LC states will be further depressed and HC states will be enhanced by negative biases, and thus the conductance difference (the light green region in Fig. 5a) gets relatively larger. In combination of electro-polarization and electrochemical gating, the intensified electric field can not only switch on-site through-space QI by structural relaxation on sub-angstrom level to maintain two conducting states, but also trigger MOs to shift their positions to modulate the switching ratios between these states. A comprehensive demonstration is provided in Supplementary Information.

To perform electrochemical gating measurement, in-situ cyclic voltammetry is firstly employed to rule out the disturbance of electrochemical reaction, and the results validate that the applied biases for electrochemical gating experiment lie within the non-Faradic region (Supplementary Note 4, Supplementary Fig. 43). The HC and LC states for $f$-Fu and $f$-Th are both spotted within accessible bias window, behaving different patterns (Fig. 5b, c, Supplementary Figs. 44–46), but $f$-Ph consistently holds sole conductance under varied circumstance (Supplementary Fig. 47). The LC states still dominate in $f$-Fu's measurement, while the HC states still become majority for $f$-Th (Supplementary Fig. 45) in dynamic break junction processes. And the HC states of both $f$-Fu and $f$-Th show symmetric conductance under positive and negative tips' biases with gradually increased conductance at higher biases. However, in terms of LC states, asymmetric conductance refers to the results that the ultralow conductance is increased with higher positive biases but decreased with enhanced negative positive biases. The HC states referring to CQI signifies comparatively symmetric transmission, which enhance conductance under higher biases. On the contrary, the LC states based on DQI comes with asymmetric transmission due to the antiresonance. As expected, the switching ratio is amplified to nearly two orders of magnitude (ca. 91) at −0.5 V in accordance with strengthened HC and suppressed LC states (Fig. 5d). Oppositely, the conductance measurement in non-polar environment without gating effect exhibit negligible fluctuation on conductance for both HC and LC states and the unwaxed tips also lead to extra anchoring arising from heterocycles (Supplementary Fig. 45).

Time-dependent conductance kinetics is performed by suspending the tip once the conducting states are detected, and holding static biases at 0.1 V in propylene carbonate (PC). The STM-BJ measurement in PC discloses that the additional HC states of $f$-Fu appear at 0.1 V but

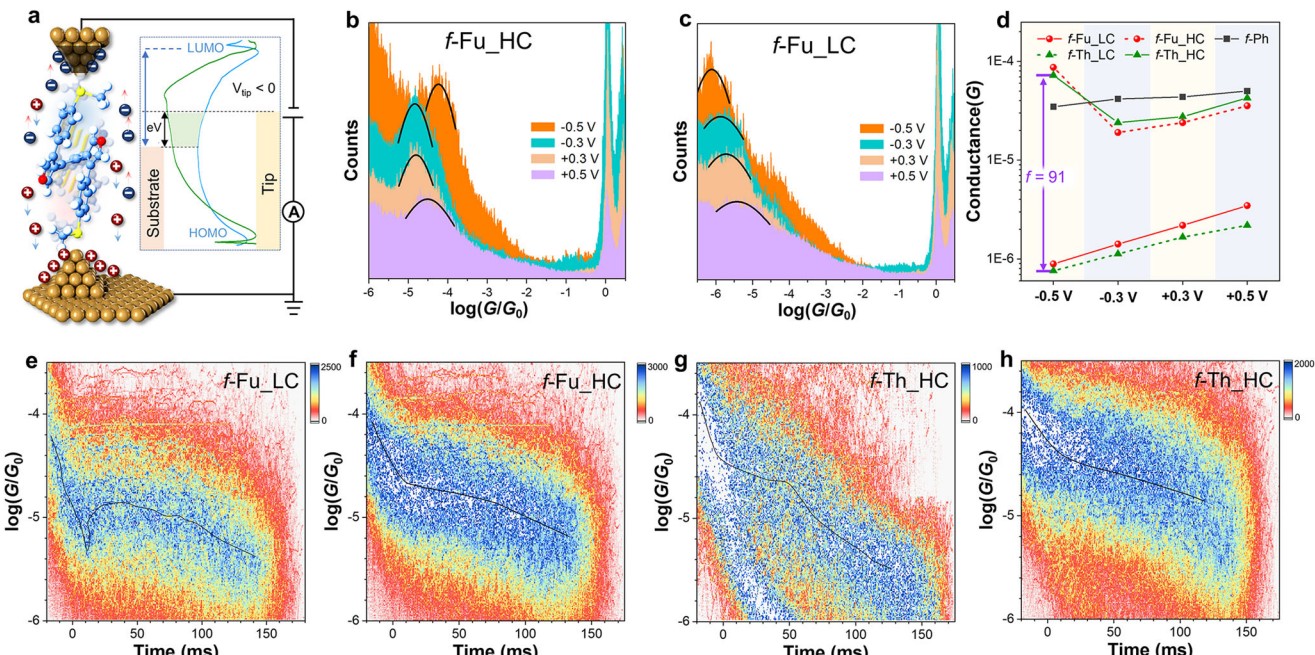

**Fig. 5 | Two-electrode electrochemical gating approach for the foldamers and suspending conductance measurement of $f$-Fu and $f$-Th in PC. a** Schematic illustration of two-electrode electrochemical gating measurement in PC with 50 mM TBAPF$_6$. The inset is energy level diagram and molecular transmission when the tip is biased negatively relative to the substrate. 1D histograms for (**b**) HC and (**c**) LC states of $f$-Fu performed by two-electrode electrochemical gating experiments at different biases. **d** Statistic conductance data of $f$-Fu, $f$-Th and $f$-Ph under different applied biases in electrochemical gating measurement with labeled switching ratios ($f$) at −0.5 V. **e** 2D conductance–displacement histograms of the LC states of $f$-Fu with visualized conductance rebound process. **f** 2D conductance–displacement histograms of the HC states of $f$-Fu. **g** 2D conductance–displacement histograms of the HC states of $f$-Th with drastic conductance drop. **h** 2D conductance–displacement histograms of the HC states of $f$-Th. The black lines refer to fitting curves of conductance.

with lower probability (Supplementary Fig. 45), while $f$-Th stays sole LC states without transformation like $f$-Ph (Supplementary Fig. 46). PC decreases the turn-on voltage for $f$-Fu by providing a polar environment. Two types of transformation are detected during the tip's suspension for $f$-Fu. The first type is that LC state is detected but switched into HC state rapidly with a rebound within 10 ms (Fig. 5e), and the second type is that HC state is detected immediately (Fig. 5f) and stays robust in the whole suspension measurement without drastic changes. Given that the molecular relaxation is a process at picosecond level, the latter situation is more consistent with the electric field-induced structural relaxation accompanied by QI switching. And the former situation of QI transformation with a delay time about 10 ms may be a comprehensive result of structural relaxation and electrode-molecule interface re-organization. Gold atoms on the surface also re-adjust themselves under electric field but with longer time constants, which may lead to sufficient structural relaxation of the molecules with adequate space and thus QI switching. This kind of re-adjustment can ensure the complete transformation of $f$-Fu but need longer time, which is not easily detected in dynamic break junction measurement. The gradually decreased HC state can be explained by electro-polarization resulting in unbalanced orbital distribution and weakened CQI. For $f$-Th, the transformation from HC state to LC state is also detectable but with a longer delay time over 50 ms (Fig. 5g). At first, the HC state maintains with a gentle slope then drastic conductance drop occurs, which is different from the preserving HC state with constant slope (Fig. 5h), indicating the transformation of QI pattern. On the basis of above discussions, the structural relaxation is less favorable for $f$-Th than $f$-Fu, therefore, a larger time constant for transformation is reasonable. Since the LC state of $f$-Th cannot be directly detected in PC at 0.1 V, the suspension measurement of the LC states is not performed. Conclusively, $f$-Th is detected with one kind of transformation with a longer responsive time over 50 ms, while $f$-Fu has two kinds of QI switching with different time constants, one is

instantaneous transformation under millisecond level and the other responds around 10 ms.

The conductance transformation under varied electric fields is also evaluated by suspending gold tip and adopting current–voltage (from the range of −1.0 to 1.0 V). After detecting molecular junction at 0.1 V, the tip suspends and applies variant bias within 150 ms. The $I - V$ curves are divided into two situations corresponding to two states found in measurement under static biases. For $f$-Fu, a small portion of junctions stay as the LC states while most of junctions switch into the HC states around $10^{-5}$ $G_0$ (Fig. 6a, d). For $f$-Th, a small portion of junctions stay as the HC states while most of junctions switch into the LC states around $10^{-5}$ $G_0$ (Fig. 6b, e). The transformation between the original states and the additional states is more effective in the process of continuous voltage variation owing to larger range of applied biases, which induces stronger polarization for the structural relaxation of the foldamer. Different from the co-existence of the HC and LC states for $f$-Fu and $f$-Th, the conductance of $f$-Ph in $I - V$ measurement continuously enhances under stronger electric field (Fig. 6c, f), probably due to the intensified through-space charge injection and enhanced electrode-molecule coupling. These phenomena further prove that the QI switching is induced by electric field rather than mechanical force, and the structural relaxation is not only responsive to stationary electric field but also varied electric fields. Besides, it turns out that once the switching is realized under low biases, the conducting state remains steady under strengthened electric fields with minor fluctuation, indicative of the potential on memory elements with low turn-on voltages. However, the switching ratios during $I - V$ sweeping is relatively smaller than those in break junction measurement, in which the HC states possess relatively lower conductance for $f$-Fu and the LC states is not suppressed enough for $f$-Th. One of the possible reasons is the suspending tips cannot provide adequate space for structural relaxation. Since the suspension is proceeded once the junctions is detected based on the setting sensitivity, the tilted molecules are not

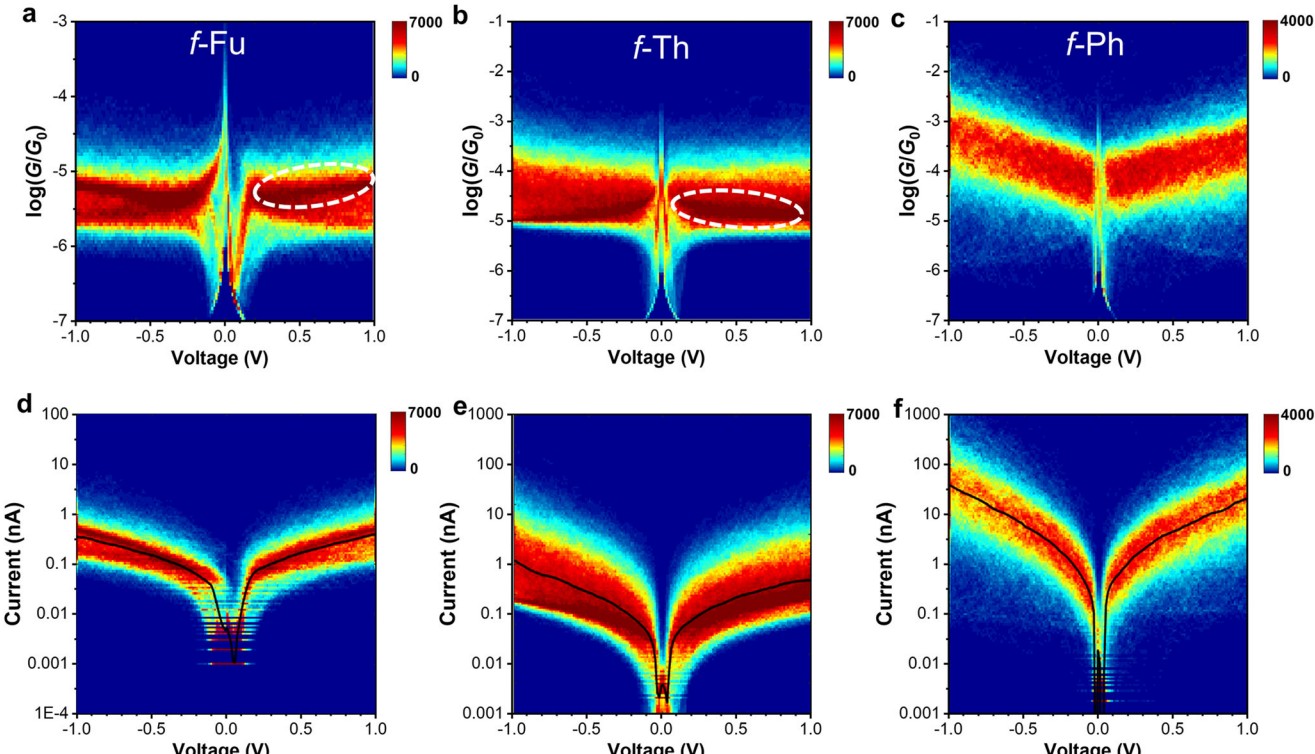

**Fig. 6 | Current–voltage (*I* − *V*) measurement for foldamers in THF/TMB (1:4, *v/v*) mixture.** 2D conductance–voltage histograms for *f*-Fu (**a**), *f*-Th (**b**) and *f*-Ph (**c**), respectively, in *I* − *V* measurement. The white dashed lines circle out the additional conducting states owing to structural reorganization under variant electric field. 2D current–voltage histograms for *f*-Fu (**d**), *f*-Th (**e**) and *f*-Ph (**f**), respectively, with black fitting curves.

able to re-adjust sufficiently. The enhanced polarization under higher biases could be additionally responsible for the lower switching ratios because the frontier orbitals are no longer delocalized on entire molecules but on one stacking arm, resulting in ineffective QI among molecular orbitals. Another reason could be the responsive time of QI switching lags behind the varying biases, which hints us to explore the time-dependent kinetics in depth for the observed switching.

## Discussion

In conclusion, in-situ voltage-triggered conductance switching on sub-angstrom level is successfully realized in foldamers *f*-Fu and *f*-Th with heterocycle−benzene stacking based on through-space interaction, and its electro-responsiveness is dominated by orbital polarization and electrostatic interaction. The introduction of furan and thiophene facilitates the formation of stable singular anti-parallel heterocycle −benzene stacking conformations. In regular conductance measurement at 0.1 V, both foldamers exhibit sole conducting state, where *f*-Fu possesses high resistance related to extremely low conductance while *f*-Th conducts with low resistance. Unexpectedly, the conducting states of *f*-Fu and *f*-Th can be partially switched on-site under higher applied biases. *f*-Fu exhibits turn-on feature with an extra HC state but *f*-Th performs turn-off feature with an additional LC state. Based on energy decomposition of non-covalent interaction, enhanced edge-to-face interaction compensates weakened stacking interaction and stabilizes folded conformations. Dispersion correction dominates intramolecular interaction but electric field mainly affects the electrostatic interaction and orbital interaction. Through-space orbital interaction determines initial CQI and DQI for *f*-Fu and *f*-Th, respectively, leading to conductance differences at 0.1 V. Then, electro-polarized orbital interaction under intensified electric field triggers structural relaxation by weakening through-space coupling, responsible for the additional conducting states induced by QI switching with the level crossing of occupied molecular orbitals. Enhanced electrostatic repulsion in *f*-Fu

assists structural relaxation and thus results in a higher transformation probability.

The switching ratio can be regulated as high as 91 by combining electro-polarization and electrochemical gating. The switching of conducting states owns higher probability in *I–V* measurement. *f*-Fu possesses two kinds of switching during the hovering of molecular junctions with different time constants, one is instantaneous transformation at millisecond level corresponding to rapid structural relaxation and the other responds around 10 ms, probably involving electrode-molecule interface re-adjustment. *f*-Th is detected with one kind of transformation with longer responsive time over 50 ms. Based on probabilistic and volatile QI switching of *f*-Fu and *f*-Th, conceptual models for TRNG and biomimetic axonic memristors are proposed. These findings provide evidences and interpretation of the electro-responsiveness of non-covalent interaction at single-molecule level, and may offer explanatory reference to the DNA and protein unfolding behaviors under electric field in the perspective of non-covalent interaction deconstruction. Moreover, it preliminarily conceptualizes the potential for natural or artificial foldamers on memristors used as TRNG to encrypted communication and biomimetic action potential to transport undecayed signals, indicating that the future development of molecular electronics may focus on the non-von Neumann architectures to surpass traditional silicon-based circuits.

The above results and discussions suggest that these tailored foldamers can be applied as voltage-triggered memory elements. But they are certainly different from non-volatile memristors as long-term resistive memory elements or synapses because their transformation among different conducting states is probabilistic and stochastic. However, the probabilistic and volatile transformation with random time constants endows the foldamers with unique advantages as random number generators or presynaptic signal transporters. On one hand, totally unpredictable random bit strings are essential for information protection and data encryption, urging the further

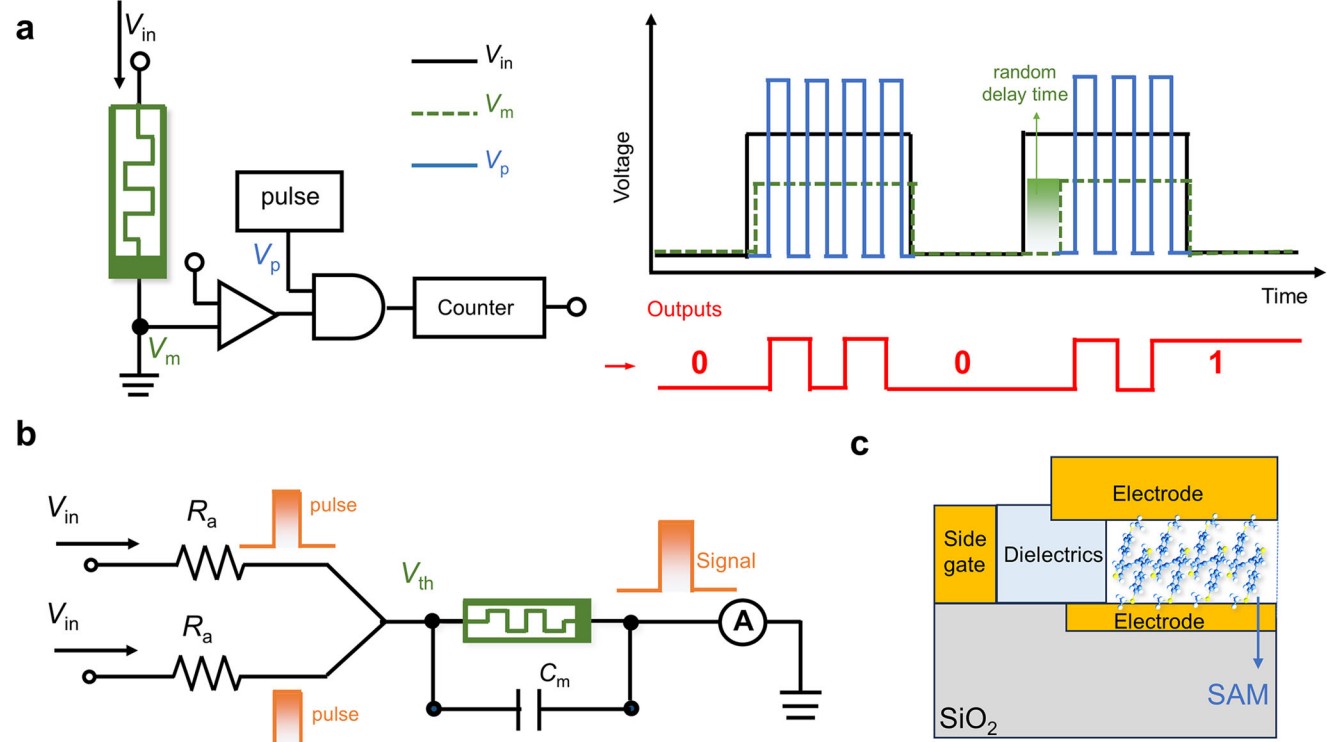

**Fig. 7 | The conceptual illustration of memory elements based on foldamers.**
**a** The TRNG model using foldamers as memristor to cooperate bits random flipping with random delay time, where the memristor is turned on with stochastic delay times, resulting in random time window for counter flipping and thus stochastic number outputs. **b** The schematic illustration of artificial neuron based on volatile foldamer memristor, where the input of voltage pulses can be translated into signal outputs if a threshold ($V_{th}$) has been reached, resulting the deformation of foldamers. **c** The schematic illustration of the half transection of metal−SAM−metal perpendicular architecture that can be integrated into devices array, where SAM refers to the self-assembly monolayer for molecules.

development of TRNG with high switching randomness. Based on probabilistic QI switching of $f$-Fu and $f$-Th, an ideal bits' array model for random outputs can be expected (Fig. 1a), but requiring perfect balance of switching to equal the probability. Another more feasible and practical solution towards TRNG may be that the foldamers are employed as volatile QI-based memristor like diffusive memristor[31] to integrate with peripheral circuity and generate stochastic delay time to ensure the randomness of counter output (Fig. 7a). For example, $f$-Fu holds two kinds of QI transformation in response to electric field with different time constants, turning LC state into HC state with lower resistance. Once an input pulse ($V_{in}$) of fixed width is applied, the memristor is turned on with stochastic delay times, resulting in random time window for counter flipping and thus stochastic number outputs. On the other hand, neuromorphic computing is in requirement of artificial neurons and synapses to achieve satisfying energy efficiency. The neuron needs to generate action potential along the circuit to transport information once a stimulus threshold is reached within a defined time interval and allows signal to decay if the interval is exceeded[13], as ideally illustrated in Fig. 1a. The feasible artificial neuron based on the foldamers receives presynaptic inputs *via* a pulsed voltage source and an equivalent synaptic resistor (Fig. 7b), and experiences conformational deformation under applied pulses, analogous to the voltage-triggered diffusive Ag filament. There will be a distinct delay time between the arrival of the voltage pulse and the rise of the output current mimicking transmission in axon, which is caused by the transformation time for the foldamer to change QI pattern once a threshold ($V_{th}$) is reached.

In addition, the large-scale solid-state integrated memristor network is highly important for the realization of neuromorphic network to proceed high throughput computing. Conventional horizontal device configuration with back-gate electrode[59] like Complementary Metal−Oxide−Semiconductor Transistor (CMOS) requires source and drain electrode etching with high precision to form metal−molecule−metal architecture, which is difficult to achieve large-scale integration. In view of this, the metal−SAM−metal perpendicular architecture[60] with side-gate electrode could be a more feasible strategy for integration (Fig. 7c). On the SiO₂ substrate with deposited bottom electrode, molecules in volatile solvent spontaneously assemble as monolayer within circular pores in a dielectric matrix surrounding by extra gate-electrode, and then the top electrode is constructed by direct metal evaporation to form a complete device. The metal nanoparticles can be introduced into the devices as well to ensure better contact between electrodes and molecules. Besides, the metal−SAM−metal architecture can avoid the problem of probabilistic switching for the foldamers by creating an ensemble average but still exploit the delay time of conductance switching, which potentially mitigates variation between devices.

All of the mentioned designing strategies imply the considerable potential of these foldamer-based molecular electronics in the areas of in-memory computing and neuromorphic computing, aiming to the breakthrough of the technology.

## Methods
### General information
All the chemicals and reagents were purchased from commercial source and used as received without further purification. ¹H and ¹³C NMR spectra were measured on a Bruker AV 500 (500 MHz) spectrometer with tetramethyl silane (TMS, $\delta = 0$). High-resolution mass spectra were recorded on a GCT premier CAB048 mass spectrometer operating in MALDI-TOF mode. Single crystal X-ray diffraction intensity data were collected on a Bruker−Nonices Smart Apex CCD diffractometer with graphite monochromated MoKα radiation.

Processing of the intensity data was carried out using the SAINT and SADABS routines, and the structure and refinement were conducted using the SHELTL suite of X-ray programs (version 6.10). High-performance liquid chromatography spectra were measured using Waters alliance e2695 separation module.

## Scanning tunneling microscopy break junction measurement

The STM-BJ setup was employed to measure the conductance of molecular junctions built between the gold tip and substrate. Gold wire (99.99%, 0.25 mm diameter) was purchased from Beijing Jiaming Platinum Nonferrous Metal Co, Ltd. for the fabrication of STM tip. All the measurements were carried out at room temperature. The gold substrates were cleaned by piranha solution before the experiments while the gold wires were carefully cleaned and annealed in a butane flame before use. The single-molecule conductance measurements were carried out using X-tech-STMBJ setup developed by Prof. Wenjing Hong's group in Xiamen University, and the data was analyzed by XMe open-source code (https://github.com/Pilab-XMU/XMe_DataAnalysis)

In STM-BJ experiment, 5 μL of the solutions containing target molecules (0.2 mM) were directly dropped on the gold substrate, and used for STM-BJ measurement. During the measurement, the gold tip was firstly controlled by a stepper motor to get to the approximate position to contact the substrate (less than 1 μm), thereafter the tip was controlled by a piezo stack with the voltage applied from 0 to 10 V. There was a feedback system in the setup, when the voltage changing of piezo stack from 0 to 10 V could not contact gold tips and substrate or break the junction to the detecting limit of our amplifier (about 1 pA), the motor would take control of the movement of the tips to offset the distances between tips and substrates. During the repeating breaking and re-connecting operation, the real-time conductance was recorded by the $I-V$ converter with a sampling rate of 20 kHz.

The gold tips used for measurements in propylene carbonate (PC) were electrochemically etched and coated with Apiezon wax. For the two-electrode gating experiments, conductance measurements of foldamers were performed with coated gold tips in 0.1 mM solution of PC with supporting electrolyte $TBAPF_6$ (50 mM). For suspending conductance experiments, the control of the piezo movement relied on the conductance at 0.1 V as sensitive feedback. The tip was suspended during the retracting process for 150 ms once the molecular junctions were detected, and conductance was recorded.

## Theoretical calculation

The molecular optimization was carried out by Gaussian16 package using M06-2x/6-311 G(d, p) basis set with DFT-D3 correction. The energy decomposition of non-covalent interaction was carried by Multifwn[37] based on molecular force field. The molecular junction was examined using the Au (111) electrodes by the Atomistic Tool Kit (ATK) software, with the optimized molecules reaching the energy minima. The molecular junction was formed by semi-finite left and right electrodes (LE and RE, respectively) and a scattering region where the molecule was settled. A scattering region comprised screening layers of LE and RE. The molecular junction was examined using the Au (111) electrodes (both LE and RE). After the optimization in Gaussian, organic molecules were inserted between the LE and RE and the whole junction configuration was optimized and the force tolerance was set to 0.05 eV Å$^{-1}$. The distance between Au and S was 3.00−3.02 Å. Then, first principles calculations were carried out to expose the electronic transport properties. In calculations, the exchange-correlation potential was approximated within the generalized gradient approximation (GGA) with Perdew−Burke−Ernzerhof (PBE) functional, for the exchange and correlation effects of the electrons. The interatomic transmission pathways at the Fermi energy were resolved with ATK software based on the calculated transmission spectra.

## Current−voltage measurement

The current−voltage measurements were carried out under the same conditions used for the conductance measurements in THF/TMB (1:4, $v/v$) mixture. The control of the piezo movement relied on the conductance at 0.1 V as sensitive feedback. The tip was suspended during the retracting process for 150 ms once the molecular junctions were detected, then the bias was continuously swept from −1.0 to +1.0 V. The data of bias, current and conductance were recorded simultaneously, and the current−voltage traces were collected with a sampling rate of 20 kHz.

## Data availability

All data supporting the findings of this study are available from the article and its Supplementary Information or available from the corresponding author. The source data underlying Fig. 2a–l, Figs. 3b, c, e, f, 4a, b, d, f, and 5b–h are provided as a Source Data file. The X-ray crystallographic coordinates for structures reported in this study have been deposited at the Cambridge Crystallographic Data Center (CCDC), under deposition numbers 2104355 (f-Fu), 2205494 (f-Th), 2205495 (f-Th-H), 2000283 (f-Ph), 2104356 (Z-Fu), 2205497 (Z-Th), 2205498 (Z-Th-2H), 2104357 (Z-Ph), 2205496 (f-Th-2H) and 2284798 (l-Th). These data can be obtained free of charge from The Cambridge Crystallographic Data Center via www.ccdc.cam.ac.uk/data_request/cif. Source data are provided with this paper.

## Code availability

The source code of the algorithms is freely available for research uses at https://github.com/JinshiLi/oPP-codes.

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

## Acknowledgements

This work was financially supported by the National Natural Science Foundation of China (22375066) and the GuangDong Basic and Applied Basic Research Foundation (2023B1515040003, 2022A1515010315, and 2019B030301003).

## Author contributions

Z.-J.Z. conceived the study and designed the experiments. J.L. and P.S. synthesized and characterized the molecules. J.L. performed STM-BJ measurement and theoretical simulation. Z.-Y.Z. and P.S. helped build the theoretical model of foldamers. J.W. performed the high-performance liquid chromatography measurement. J.L. and Z.-J.Z. wrote and revised the manuscript. Z.-J.Z. and B.Z.T. supervised the project. All authors discussed the results and commented on the manuscript.

## Competing interests

The authors declare no competing interests.
