## [Peer Review File · Nature Communications]

In-situ electro-responsive through-space coupling enabling foldamers as volatile memory elementsREVIEWER COMMENTS

Reviewer #1 (Remarks to the Author):

In this manuscript, the authors design tailored foldamers with furan-benzene stacking (f-Fu) and thiophene-benzene stacking (f-Th) to decipher electro-responsive, which can achieve volatile memory behaviors via quantum interference switching in single-molecule junctions. The prepared f-Fu can realize complete transformation, while f-Th suffers incomplete switching and longer responsive time. This is similar to voltage-gated channels in axons and has potential applications on non-von Neumann computing. In this work, the authors achieve relatively high switching ratios of 91 for f-Fu by additional electrochemical gating, which provides valuable evidence and interpretation of the electro-responsiveness of non-covalent interaction at single-molecule level and offer novel design strategies to true random number generator and axon-like signal transmission. The authors have done abundant synthesis, simulation, experimental and electrical characterizations in this work to demonstrate and explain the results. The idea is also new and very interesting. I suggest a minor revision.

Some comments are listed as follows to further improve this manuscript.

1. "So" appears many times as a connection word between two sentences. It is too colloquial and suggested to be revised.
2. "adopting current-voltage (from the range of -1.0 V to 1.0 V." lacks of ")".
3. Schematic illustration of the setup and electrolyte for the two-electrode electrochemical gating measurement is suggested to be added in supporting information for easier reading and inspiration. What's the selection criteria for the electrolyte or solution environment?
4. What's the raw data format (only curve format) for the coordinates of counts vs $\log(G/G_0)$ in figure 2, figure 3, and figure 5? It's suggested to be added in supporting or response letters. What's the unit for "counts"?
5. The authors are recommended to explain the reason to put figure 3 (control experiments) in the main text instead of supporting info.
6. Add the scale bar for figure 6 d, e, f.
7. Mark f-Fu and f-Th in figure 1a.
8. How about the purity of the synthesized f-Fu and f-Th? How about its influence on the electro-responsive properties? This part should be explained.
9. As the demonstration of high switching ratio need electrochemical measurement, how to achieve solid-state devices based on the proposed foldamers? How to realize the integration with state-of-art technology? This part should be carefully discussed in the conclusion part with perspective.

Reviewer #2 (Remarks to the Author):

The paper by Zhao and co-workers reported a kind of voltage-driven in-situ molecule switches based on ortho-linked aromatic foldamers and proposed a design strategy of non-von Neumann electronics based on these foldamers stochastic dynamic switching. The inherent switching mechanism was explained as sub-angstrom stacking malposition due to molecular polarization and thus electrostatic potential re-distribution, which resulted in quantum interference switching. The conceptual design of random number generator and axon-like signal transmission presented in this work is quite novel and enlightening for molecular electronics field, suggesting a new and important direction to be developed in future. The investigation is well conducted, the experimental results and scientific

discussions are sufficient to support the conclusions. The reviewer suggests this work can be accepted in Nature Communications after addressing the following minor issues.

1. The authors claim that the HC and LC states for f-Fu and f-Th appear independently in break junction measurement, but the total junction formation probability of HC and LC states for f-Fu is over 100%. Additional explanation is needed for this probability.
2. The manuscript put forward a few potential applications based on non-Neumann architectures, which are very interesting and meaningful. In view of that these designs are quite new to molecular electronics, a more proper and detailed description of feasible true random number generator and axon-like transmission should be supplemented.
3. The previous research has mentioned the mechanical elongation of ortho-phenylene derivatives, is it possible that the LC states actually stem from the unfolded conformer? How to exclude the possibility of mechanical elongation caused by moving tip?
4. Why the energy of orbital interaction can't be calculated? The detailed calculated methods should be provided.
5. The foldamers' MPSH at 0.2 V and molecular orbital distribution under 0.0002 a.u. are different. Once the orbitals are polarized, will the quantum interference pattern based on through-space interaction still dominate molecular conductance?
6. Some minor conductance peaks in the Z-shaped molecules are not explained. Z-Th is believed to have multiple conductance since the thiophene is able to anchor with electrodes. How can the authors ascribe the peaks around 10^{-5} G Ω to the contacting conformers with two thiomethyl groups anchored to the electrodes?

Reviewer #3 (Remarks to the Author):

In this work, Li et al. discovered and explored voltage-gated structural relaxation in foldamer-based single-molecule junctions, which can induce additional conducting states and have the potential to act as volatile memory elements as they claimed. The way electric fields manipulate intramolecular through-space interactions is interesting, and the authors dig deep into the conductance switching mechanism. But there are several issues that need to be addressed before this manuscript can be considered for publication in Nature Communications.

1. The "turn on" in f-Fu molecule and "turn off" in f-Th molecule are probabilistic events, in other words, they cannot work like conventional volatile memristors whose conductive states can be precisely controlled by the applied voltages. Is it possible to manipulate the HC and LC states as desired?
2. The on-off ratios achieved by pure electro-polarization are about 14 at 0.5 V (Fig. 2j), which seems to be in the same range realized by the electro-polarization with additional electrochemical gating at 0.5 V (Fig. 5d). On the other hand, switching ratio of 91 can be obtained with the help of electrochemical gating at -0.5 V, but the corresponding data at -0.3 and -0.5 V in Fig. 2j are missing. To elucidate the impact of electrochemical gating on the switching performance, it is necessary to compare under a similar situation. And it is also recommended to clarify in detail how the ions modulate the switching property.
3. As illustrated in Fig. 1a (bottom), the authors propose that their molecular junctions based on heterocycle-benzene stacking foldamers have the potential to be used as true random number generators and axon-like voltage-gated channels, which greatly raised my expectations. However, it is still a conceptual scenario at the end of this work. Fig. 7e and f are just the extended schematic illustration of Fig. 1a (bottom). It will be meaningful if the

authors could provide some demos to demonstrate their ideas. Otherwise, this part should be carefully reorganized.

4. The authors use “ineffective conductance” and “effective charge transport” to describe the relatively low and high conductance, respectively. It seems kind of weird because, in fact, the difference in conductance between Fig. 2b and Fig. 2f is not large enough to use “ineffective” and “effective”.

5. What’s the definition of “start-up voltage”? Is there any connection with the SET/RESET voltages in memristors? By the way, can these molecular junctions start up around -0.2 V?

6. In Fig. 7a-d, the QI transformation can be as fast as 10 ms and it has a fast response time. Is it possible to realize repeatable HC-LC switching in the molecular junctions reported here? And how about the responsive time during this cycling?

7. The authors prepared a series of controlled molecules (such as foldamers with one SMe anchor, without SMe anchor, unstacked, etc.) to elucidate the basic switching mechanism. However, some phenomena are not clearly explained. For example, the two conducting states that appeared in the f-Th-H junction are attributed to an assumption. On the other hand, can extramolecular π -stacking occur between furan-benzene, thiophene-benzene, or benzene-benzene in these molecular junctions?

8. There are some errors that need to be corrected, including but not limited to:

1) Statistic conductance data of f-Ph under different applied biases are not presented in Fig. 2j as described in the legend.

2) According to the insert in Fig.2f, the observed junction length of f-Th is 1.28 nm after adding 0.5 nm snap-back distance, instead of 1.25 nm. Please check this value.

3) At the end of Page 4, “But the minor LC states of f-Fu own higher probabilities around 40% than the minor HC states of f-Th around 30%...”, are the minor states of f-Fu and f-Th HC and LC, respectively?

4) The legend of Fig. 4a (bottom panel) is missing.

5) In Page 11, the figure number should be Fig. 6, instead of Fig. 5.

6) Fig. 7a-d are not “2D conductance-displacement histograms”, and the corresponding color bars should be provided in these figures.

Response to Reviewer #1

Some comments are listed as follows to further improve this manuscript.

1. “So” appears many times as a connection word between two sentences. It is too colloquial and suggested to be revised.

Answer: We thank reviewer’s good suggestions. The expression is carefully revised as more formal form.

2. “adopting current–voltage (from the range of –1.0 V to 1.0 V.” lacks of “”).

Answer: We thank reviewer’s detailed suggestions. This error has been corrected. And the entire manuscript is carefully checked.

3. Schematic illustration of the setup and electrolyte for the two-electrode electrochemical gating measurement is suggested to be added in supporting information for easier reading and inspiration. What’s the selection criteria for the electrolyte or solution environment?

Answer: We thank reviewer’s good suggestions. The selected solvent needs to be polar to provide certain solubility for supporting electrolyte and form an ionic environment. And the solvent is preferred to have a large bias window to prevent polarization or re-dox reaction during the gating experiment. Acetonitrile is too volatile to maintain ionic concentration during the whole measurement lasted for a few hours. *N,N*-Dimethylformamide and dimethyl sulfoxide have strong absorption of water in ambient environment, which is not good for the measurement. Therefore, propylene carbonate is selected as the testing solvent. The electrolyte is required to be inert and does not react with solvents or samples. Given the addition of TBAPF₆ would not affect the current baseline of the testing system, TBAPF₆ is selected as supporting electrolyte.

The purpose of electrochemical control in two-electrode electrochemical gating experiment based on STM-BJ is to form an asymmetric, bias-dependent electric double layer in polar environment to adjust the relative alignment between the Fermi level and molecular levels. In polar solvent, with transferable ions, electric double layers at the tip and substrate are formed by ions’ movement to screen out the electric field due to charges on the metal, influencing the electrostatic environment around the junction. Specifically, the coated tip is capable of forming a denser double layer with almost single-metal-atom exposition to the solvent, resulting in electrostatic asymmetry and bias polarity-dependent shift of the molecular resonance energy. The detailed description is complemented in Supplementary Information.

4. What’s the raw data format (only curve format) for the coordinates of counts vs $\log(G/G_0)$ in figure 2, figure 3, and figure 5? It’s suggested to be added in supporting or response letters. What’s the unit for “counts”?

Answer: We thank reviewer’s good suggestions. In the measurement of single-molecule conductance, thousands of individual single conductance-displacement traces are recorded in each experiment and apply statistical approach to determine the most probable conductance and the stretching distance (as shown in Fig. R1, cited from *Beilstein J. Nanotechnol.* **2**, 699(2011)). The 1D conductance histograms were constructed by collecting individual traces with a bin size of 1100 for $\log(G/G_0)$ from –10 to +1, and 1000 for the distance between the tip and substrate (Δz) from –0.5 to 3 nm. The conductance distribution was extracted by calculating the data density in each bin. The peak shift in a conductance histogram was determined by Gaussian fitting, which represents the most probable molecular

conductance. The 2D conductance-displacement histograms were plotted by overlapping each individual trace with a bin size of 1100 for $\log(G/G_0)$ from -10 to $+1$, and 1000 for Δz from -0.5 to 3 nm. All the traces are aligned with a relative zero point ($\Delta z = 0$) at $G = 0.5 G_0$. Then the 2D conductance distribution versus the relative distance was constructed by the data counts in each bin. To construct the displacement distribution histograms, firstly the relative stretching distance, Δz , was determined from the position where the conductance is $0.5 G_0$ (after the rupture of the gold-gold atomic break at G_0), to the molecular conductance region, just before the end of molecular plateau. The peak represents the most probable plateau length. The raw data format is the single conductance-displacement traces. The “counts” refer to the counting of recorded data, reflecting the most probable conductance for single-molecule junction. These detailed introductions are complemented in Supplementary Information.

Fig. R1 **a** Traces constructed from the slopes of individual I - V curves as recorded simultaneously during a slow stretching half cycle, with (red traces) and without (blue traces) the formation of a molecular junction. **b** Conductance histogram as constructed from the data points plotted in panel **a**

5. The authors are recommended to explain the reason to put figure 3 (control experiments) in the main text instead of supporting info.

Answer: We thank reviewer’s good question. When it comes to multiple conductance in the measurement for molecular junctions, multi-anchors with different contacting geometry are always the initial assumption for chemists, especially for the molecules with thiophenes. And the linkers changing from benzene to furan and thiophene may also raise the confusion of the origin of the conducting difference. Does the conductance difference among f -Ph, f -Fu and f -Th come from the alteration of through-bond connection or through-space interaction? Figure 3 in main text illustrates the conductance behaviors of the control molecules and provides evidences that the conductance difference among f -Ph, f -Fu and f -Th does not stem from the through-bond pathway or multi-anchoring, subsequently revealing the difference of through-space interaction among these foldamers. It is a part of argumentation to exclude the common possibility, which is important. And these control experiments are accompanied with more additional measurements which have been already provided in Supplementary Information. If the whole control experiment part is just provided in Supplementary Information, necessary extra discussions might be ignored by the readers.

6. Add the scale bar for figure 6 d, e, f.

Answer: We thank reviewer’s good suggestions. The scale bars are complemented in Fig 6d-f.

7. Mark f-Fu and f-Th in figure 1a.

Answer: We thank reviewer's good suggestions. The molecular names are complemented in Fig. 1a.

8. How about the purity of the synthesized f-Fu and f-Th? How about its influence on the electro-responsive properties? This part should be explained.

Answer: We thank reviewer's good suggestions. The purity of the studied molecules in this manuscript has been verified by high-performance liquid chromatography and the corresponding spectra are provided in Supplementary Fig. 4. No detectable impurity was found in the tested samples, so it is reasonable to believe that the switching behaviors for the foldamers do not come from the impurity.

9. As the demonstration of high switching ratio need electrochemical measurement, how to achieve solid-state devices based on the proposed foldamers? How to realize the integration with state-of-art technology? This part should be carefully discussed in the conclusion part with perspective.

Answer: We thank reviewer's good suggestions. There are two major solutions for the accomplishment of high switching ratios based on electrochemical modulation. For starters, molecular engineering is a strategy to insert inherent ionic environment for the replacement of external ionic environment. For example, exchanging furan or thiophene into pyridinium salt or oxidized pyrrole can introduce counter anion to provide ionic environment in solid state. The migration of counter anion may also possess gating effect for the working foldamers without gating electrode, which are in need of further exploration and verification. The other alternative is to construct a solid gating electrode. Conventional horizontal device configuration with back-gate electrode like CMOS requires source and drain electrode etching with high precision to form metal–molecule–metal architecture, which is difficult to achieve integration. We think that the metal–SAM–metal perpendicular architecture with side-gate electrode constructed by self-assembly monolayer is a more feasible strategy for integration (as shown in Fig. 7c in the revised manuscript), taking previous report on large-scale integration of molecular devices for reference (*Nature*, **559**, 232-235(2018)). On the SiO₂ substrate with deposited bottom electrode, molecules spontaneously assemble as monolayer within circular pores in a dielectric matrix surrounding by extra gate-electrode, and then the top electrode is constructed by direct metal evaporation to form a complete device. The metal nanoparticles can be introduced into the devices as well to ensure better contact between electrodes and molecules. Besides, the metal–SAM–metal architecture can avoid the problem of probabilistic switching for the foldamers by creating an ensemble average but still exploit the delay time of conductance switching, which potentially mitigates variation between devices. The detailed discussions for the realization of integrated solid-state devices are complemented as perspective in the section of Summary and Outlook in the revised manuscript.

Response to Reviewer #2

1. The authors claim that the HC and LC states for f-Fu and f-Th appear independently in break junction measurement, but the total junction formation probability of HC and LC states for f-Fu is over 100%. Additional explanation is needed for this probability.

Answer: We thank reviewer's good suggestions. The junction formation probability is analyzed based on an auto-

classification algorithm of the spectral clustering. Once a molecular plateau in the preset conductance range (such as $10^{-4.2} \sim 10^{-5.0} G_0$ for HC states or $10^{-5.0} \sim 10^{-5.8} G_0$ for LC states) is detected, the corresponding conductance-displacement traces would be counted as valid data to calculate the junction formation probability. Given the structural transformation is probabilistic in these experiments, it is reasonable that the unrelaxed conformation attached to the electrodes first and then achieve *in-situ* transformation to another conducting states. In that case, two contacting geometries corresponding to two conducting states can be observed simultaneously in one stretching process. When calculating junction formation probability, this kind of traces would be double-counted leading to the seemingly exaggerated junction formation probability over 100%.

2. The manuscript put forward a few potential applications based on non-Neumann architectures, which are very interesting and meaningful. In view of that these designs are quite new to molecular electronics; a more proper and detailed description of feasible true random number generator and axon-like transmission should be supplemented.

Answer: We thank reviewer's good suggestions. The initial inspiration for foldamers as volatile memristors comes from the generality between diffusive memristors based on the diffusion of silver filament and the voltage triggered elongation of foldamers. One of the feasible and practical solution towards TRNG may be that foldamers are adopted as volatile QI-based memristor like diffusive memristor to integrate with peripheral circuitry and generate stochastic delay time to ensure the randomness of counter output (Fig. 7a in the revised manuscript). For example, *f*-Fu possesses two kinds of QI transformation in response to electric field with different time constants, turning LC state into HC state with lower resistance. Once an input pulse of fixed width is applied, the memristor is turned on with stochastic delay times, resulting in random time window for counter flipping and thus stochastic number outputs. As for neuron, it needs to generate action potential along the axon to transport information once a stimulus threshold is reached within a defined time interval and allow signal to decay if the interval is exceeded. Therefore, the artificial neuron based on foldamers receives presynaptic inputs via a pulsed voltage source and an equivalent synaptic resistor (Fig. 7b in the revised manuscript) mimicking the biological temporal behavior. There would be a distinct delay time between the arrival of the voltage pulse and the rise of the output current, which is caused by the transformation time for the foldamer to change QI pattern once a threshold is reached. The detailed discussions for the realization of a true random number generator and axon-like transmission is complemented as perspective in the section of Summary and Outlook in the revised manuscript.

3. The previous research has mentioned the mechanical elongation of ortho-phenylene derivatives, is it possible that the LC states actually stem from the unfolded conformer? How to exclude the possibility of mechanical elongation caused by moving tip?

Answer: We thank reviewer's good suggestions. Considering the piezo rates has a profound effect on the junction formation probability, the piezo rates applied in scanning tunneling microscope break junction (STM-BJ) measurement need to be chosen discreetly. The piezo rates of 20, 30 and 40 nm s⁻¹ are applied in conductance measurement of *f*-Fu, *f*-Th and *f*-Ph in the mixture of tetrahydrofuran and mesitylene. Only one-distinct conductance peak can be found under different piezo rates in the measurement of *f*-Fu at 0.1 V. For *f*-Fu, the LC state possesses short statistical junction length than the HC states, indicating shorter molecular length. And the unfolded conformation owns the longest molecular length comparing to other conformations. Therefore, the possibility for LC state of *f*-Fu stemming from the unfolded conformation can be excluded. As for *f*-Th, the junction length for LC states is slightly longer than the HC states around 0.3 Å, while the unfolded conformation is longer than the folded conformation over 1 Å. Therefore, the LC state for *f*-Th is more likely to stem from a folded conformation with slight

deformation rather than the completely unfolded conformation.

To further verify that the switching of conducting states does not originate from the mechanical force, conductance measurement at 0.2 V under a piezo rate of 40 nm s⁻¹ is also performed. Both HC states and LC states for *f*-Fu and *f*-Th are still observable. When lowering the piezo rates at 0.1 V, switching does not happen but just one conductance peak is detected for these foldamers. These results comprehensively illustrate that the switching is voltage-induced than mechanical force-induced. The above discussion is complemented in Supplementary Information.

4. Why the energy of orbital interaction can't be calculated? The detailed calculated methods should be provided.

Answer: We thank reviewer's good suggestions. The present energy decomposition calculation of through-space interaction including the orbital interaction (such as Symmetry-Adapted Perturbation Theory, SAPT) needs to divide the ensemble wavefunction into the addition of segments' wavefunctions to quantify the orbital interaction. However, segmentation of the ensemble system is more useful in the analysis of intermolecular through-space interaction among multiple molecules without consideration of radicals. For intramolecular through-space interaction, the research object is usually one single molecule. After segmentation, it is inevitable to yield molecular fragment with radicals. The introduction of radical fragment may cause spin pollution during the demonstration of energy decomposition leading to inaccurate results. Most of energy decomposition analyses do not support the analysis of intramolecular through-space interaction. The energy decomposition based on force field can provide the intramolecular energy decomposition. Since it does not calculate with the basis of wavefunction but the molecular force field, the information of orbital interaction is missing. But the orbital interaction can be visualized in the molecular orbital distribution. The detailed calculating methods have been complemented in Supplementary Information.

5. The foldamers' MPSH at 0.2 V and molecular orbital distribution under 0.0002 a.u. are different. Once the orbitals are polarized, will the quantum interference pattern based on through-space interaction still dominate molecular conductance?

Answer: We thank reviewer's good questions. The MPSH is calculated from the device density of states based on device configuration with periodic boundary condition and the molecular orbital distribution is calculated from the electronic wavefunction of isolated molecules. These two results are obtained from two different calculating methods, so the differences in exhibiting electronic distribution is reasonable. But both of MPSH and molecular orbital share a same tendency that molecular orbitals would be polarized under electric field, convincing the reliability of our hypothesis that the electro-polarization is the reason of conductance switching in foldamer system. It is assumed that quantum interference pattern based on through-space interaction is incapable of dominating molecular conducting any longer under strong electric field. Once the orbitals are strongly polarized under strong electric field, the energy levels of two stacked arms will not match and turn into isolation because of the Stark effect, so that their orbital cannot couple to form an effective continuous pathway. The isolation of energy levels may cause ineffective conductance and disable the conductance switching, which may result in rectification with unbalanced conductance. But in the current-voltage measurement for *f*-Fu and *f*-Th, no rectification is observed, illustrating that electro-polarization is not strong enough to isolate the coupling of stacked arms under the applied voltage (≤ 0.5 V). As the coupling of stacked arms still exists, the quantum interference pattern still dominates the molecular conductance in the conductance measurement in this work.

6. Some minor conductance peaks in the Z-shaped molecules are not explained. Z-Th is believed to have multiple conductance since the thiophene is able to anchor with electrodes. How can the authors ascribe the peaks around $10^{-5} G_0$ to the contacting conformers with two thiomethyl groups anchored to the electrodes?

Answer: We thank reviewer's good suggestions. There are three conducting states can be found during the STM-BJ measurement for Z-Th, which are $10^{-3.72}$, $10^{-4.15}$ and $10^{-5.00} G_0$, respectively. Since the conductance of linear *l*-Th is $10^{-2.72} G_0$, those three kinds of result are not likely to correspond to the device contacting geometry that one arm of Z-Th attaches to the electrodes. Based on the molecular length calculated from plateau lengths and simulated transmission results, it is concluded that the conductance of $10^{-3.72}$ originates from two thiophene anchored junction as shown in Supplementary Fig. 16f. The linear and planar structure is indicative of better conjugation. The similar conductance result of Z-Th-2H at $10^{-3.82} G_0$ further verifies the conclusion above.

The absence of $10^{-5} G_0$ in the measurement of Z-Th-H indicates that the conductance of $10^{-5} G_0$ originates from the two SME anchored junction, which is marked as "Z-Th*" and compare to foldamers as control experiment. Excluding all other possibility, it is believed that the conductance of $10^{-4.15} G_0$ corresponds to the conductance stemming from one SME and one thiophene anchored junction, and the referring thiophene is far away from SME group rather than directly covalently bonded with thiomethyl benzene. The contacting geometry has been shown in Supplementary Fig. 16f. The corresponding discussions have been provided in Supplementary Information.

Response to Reviewer #3

1. The "turn on" in *f*-Fu molecule and "turn off" in *f*-Th molecule are probabilistic events, in other words, they cannot work like conventional volatile memristors whose conductive states can be precisely controlled by the applied voltages. Is it possible to manipulate the HC and LC states as desired?

Answer: We thank reviewer's good questions. For the device configuration of STM-BJ measurement, one of the electrodes is the gold tip that melted as sphere by butane and the other is a flat gold substrate. In this regard, the actual electric field between two electrodes are analogous to spherical electric field with anisotropy rather than uniform electric field. The dispersing molecules would possess different degree of polarization owing to the uneven electric field, leading to incomplete switching of molecular conformation. Given that STM-BJ measurement is a dynamic process with high sensitivity, both polarized and unpolarized molecules are able to be trapped by the electrode to form molecular junction. Therefore, it is difficult to manipulate the switching between HC and LC states as desired in dynamic break junction measurement. However, during the suspension of tip, a complete transformation from LC state to HC state for *f*-Fu can be observed. Because the electric field is strongest along the direction of tip lifting, molecule can be effectively polarized and relaxed in static junctions. To conclude, the break junction technique is convenient to explore the feature of molecular junction but fail to achieve complete transformation, while static junction possesses the potential to accomplish desired transformation but still need further development of related on-chip devices. The design strategy of solid devices is provided as perspective in the revised manuscript, and the following work focusing on practical device demo requires further cooperation with other team and in-depth exploration on different molecular system.

2. The on-off ratios achieved by pure electro-polarization are about 14 at 0.5 V (Fig. 2j), which seems to be in the same range realized by the electro-polarization with additional electrochemical gating at 0.5 V (Fig. 5d). On the other hand, switching ratio of 91 can be obtained with the help of electrochemical gating at -0.5 V, but the corresponding data at -0.3 and -0.5 V in Fig. 2j are missing. To elucidate the impact of electrochemical gating on the switching

performance, it is necessary to compare under a similar situation. And it is also recommended to clarify in detail how the ions modulate the switching property.

Answer: We thank reviewer's good suggestions. The conductance measurements under -0.3 and -0.5 V in non-polar solution are also performed and their statistical data are shown in Fig. R2 with comparison to the two-electrode electrochemical gating experiment. It can be seen that, in non-polar environment, the conductance for the HC states and LC states of foldamers just fluctuate within a small range without distinct changing tendency. Oppositely, in polar environment, the LC states of foldamers exhibit asymmetric conductance while the HC states of foldamers exhibit relatively symmetric conductance under different direction of applied voltages.

The purpose of electrochemical control in two-electrode electrochemical gating experiment based on STM-BJ is to form an asymmetric, bias-dependent electric double layer in ionic environment to adjust the relative alignment between the Fermi level and molecular levels. In polar solvent with transferable ions, electric double layers at the tip and substrate are formed by ions' movement to screen out the electric field due to charges on the metal, influencing the electrostatic environment around the junction. Specifically, the coated tip is capable of forming a denser double layer with almost single-metal-atom exposition to the solvent, resulting in electrostatic asymmetry and bias polarity-dependent shift of the molecular resonance energy. The detailed mechanistic demonstration for the change of conducting behaviors in ionic environment has been provided in Supplementary Information.

Fig. R2 Statistic data of conductance (a) and junction formation probability (b) for conductance measurement under negative and positive biases in THF: TMB (1:4, v/v). Statistic data of conductance (c) and junction formation probability (d) for conductance measurement under negative and positive biases in PC.

3. As illustrated in Fig. 1a (bottom), the authors propose that their molecular junctions based on heterocycle-benzene stacking foldamers have the potential to be used as true random number generators and axon-like voltage-gated channels, which greatly raised my expectations. However, it is still a conceptual scenario at the end of this work. Fig. 7e and f are just the extended schematic illustration of Fig. 1a (bottom). It will be meaningful if the authors could provide some demos to demonstrate their ideas. Otherwise, this part should be carefully reorganized.

Answer: We thank reviewer's good suggestions. Our present work focuses more on the development of materials, the realization of practical devices is a brand-new area for us. It needs other experienced cooperators to carry out relevant researches on practical demos, which is another research project we desired to proceed in the future. The inspiring questions from reviewers have encouraged us to come up with more feasible strategy towards device fabrication. The initial inspiration for foldamers as volatile memristors comes from the generality between diffusive memristors based on the diffusion of silver filament and the voltage triggered elongation of foldamers. So, based on the current construction of diffusive memristors for non-von Neumann architecture, we bring about our designing strategy for molecular volatile memory element as illustrated in Fig. 7a and 7b in order to arouse the attention and discussion on molecular electronics as the state-of-the-art technique like neuromorphic network rather than traditional devices. Fig. 1a is more like an ideal concept for volatile memory element while Fig. 7 is inclined to design molecular electronics in a more practical way and exhibit actual potential for molecular electronics on non-von Neumann architecture. Undeniably, we still have difficulty in achieve practical demos, so this part is reorganized as perspective in the section of summary and outlook.

4. The authors use "ineffective conductance" and "effective charge transport" to describe the relatively low and high conductance, respectively. It seems kind of weird because, in fact, the difference in conductance between Fig. 2b and Fig. 2f is not large enough to use "ineffective" and "effective".

Answer: We thank reviewer's good suggestions. The word of "ineffective" and "effective" are comparative usage to make comparison with the conductance for foldamers, which may cause misunderstanding. All the usage of "effective" and "ineffective" is modified in the manuscript, using more precise expression as "higher" and "lower" conductance.

5. What's the definition of "start-up voltage"? Is there any connection with the SET/RESET voltages in memristors? By the way, can these molecular junctions start up around -0.2 V?

Answer: We thank reviewer's good suggestions. The start-up voltage refers to the voltage that can induce quantum interference switching for *f*-Fu and *f*-Th. In our experiment based on STM-BJ, 0.2 V is the voltage that can induce conductance switching and thus result in the co-existence of two conducting states. When these foldamers is applied as volatile memristors, the start-up voltage as 0.2 V would be the SET voltage in memristors to induce conformational elongation to switch the molecular conductance and RESET voltage can be 0.05 or 0.1 V that cannot induce sufficient conformational relaxation to switch the conductance.

The conductance measurement was also performed under -0.2 V. The switching from HC state to LC state for *f*-Th and from LC states to HC states for *f*-Fu are also detectable under -0.2 V, but with lower junction formation probability (Fig. R3). It can be explained by the enhanced resistance between molecular anchor (thiomethyl group) and the tip with negative electric potential. The detailed discussion is complemented in the Supplementary Information.

Fig. R3 2D histograms for the conductance measurement of *f*-Fu and *f*-Th under -0.2 V.

6. In Fig. 7a-d, the QI transformation can be as fast as 10 ms and it has a fast response time. Is it possible to realize repeatable HC-LC switching in the molecular junctions reported here? And how about the responsive time during this cycling?

Answer: We thank reviewer's good suggestions. Because our present equipment performing STM-BJ technique does not support to circularly change the applied voltage during the tip's suspension yet, it is incapable to perform repeatable HC-LC switching. The responsive time of 10 ms actually cannot meet the demand of the practical usage with microsecond responsive time. As we discussed in the manuscript, the conformation change of molecules is much faster within picosecond beyond the equipment's detection. What we observed within the range of 10 ms may owe to the extra dynamics of electrode migration. Therefore, the precise observation of switching time for HC and LC states induced by the conformational change of molecules require higher precision of equipment and further development of software programming to apply circular experiment.

Fig. R4 Conductance measurements of *f*-Th-H and *l*-Th. **a** 2D histograms of conductance measurement for *f*-Th-H. **b** Schematic illustration of contacting device configuration corresponding to two different conducting states for *f*-Th-H. **c** 1D histograms of conductance measurement in THF: TMB (1:4, v/v) for *l*-Th. **d** 2D histograms of conductance measurement for *l*-Th.

7. The authors prepared a series of controlled molecules (such as foldamers with one SMe anchor, without SMe anchor, unstacked, etc.) to elucidate the basic switching mechanism. However, some phenomena are not clearly explained. For example, the two conducting states that appeared in the *f*-Th-H junction are attributed to an assumption. On the other hand, can extra molecular π -stacking occur between furan-benzene, thiophene-benzene, or benzene-benzene in these molecular junctions?

Answer: We thank reviewer's good suggestions. The extra experiments have been carried out and the additional discussions have been provided in Supplementary Information.

(1) *f*-Th-H exhibits two blurry conductance peaks at $10^{-2.97}$ and $10^{-5.07} G_0$, with junction lengths of 0.56 nm and 0.108 nm after calibration, respectively. To decipher the origin of these states, extra linear *l*-Th molecule is applied for extra conductance measurement for comparison. *l*-Th has been reported previously (*Angew. Chem. Int. Ed.* **59**, 3280–3286(2019)) exhibiting one relatively higher conductance with shorter molecular length referring to the conductance of monomer and the other relatively lower conductance with longer molecular length referring to the conductance of dimer. In that case, it is probable to presume the conductance state at $10^{-2.97} G_0$ for *f*-Th-H is attributed to the contacting configuration that thiomethyl group (SMe) attaches to one electrode while thiophene is attached to the other. Considering the stacked arms provide certain steric hindrance and the distance between SMe and the sulfur atoms in thiophene on single through-bond connected arm is slightly longer than 0.56 nm (around 0.8 nm), thiophene probably couples to the electrode as an ensemble aromatic ring. As for the peak at $10^{-5.07} G_0$, it may come from the through-space charge injection *via* terminal benzene (Fig. R4b) in the consideration of the similarity to the dimer conductance peak for *l*-Th. The absence of one SMe is responsible

for the weakened conductivity and shorter statistical molecular length. In conclusion, two blurry conductance peaks around 10^{-3} and $10^{-5} G_0$ are attributed to the anchoring between SMe and thiophene and through-space charge injection *via* terminal benzene, respectively.

- (2) On one hand, even in relatively closer packing for single crystal than in solution (Fig. R5), there is no intermolecular π -stacking in crystal cell as the closest distances between inter-aromatic rings are longer than 3.5 Å and just a few negligible C–H $\cdots\pi$ interaction. The folded and twisted structure is helpful to inhibit intermolecular π -stacking. Unlike the planar *l*-Th with a very small torsion angle of 3.72° between thiophene and benzene, the stacked arms in *f*-Fu and *f*-Th hold torsion angles larger than 20°, which are not favorable for the formation of intermolecular through-space interaction.

Additional conductance measurements are also designed for deepened discussion. The intermolecular stacking usually depends on the molecular concentration, and the probability of stacking increases with the increasing concentration. But in conductance experiments with a higher concentration (1 mM), it is found that both heterocycle-benzene stacking foldamers remain single conductance peak at 0.1 V as well as co-existence of dual-states at 0.3 V (Fig. R5), and the HC state for *f*-Fu and LC state for *f*-Th stay as minor states, respectively. The only difference is that the peak at $10^{-3} G_0$ becomes more distinct in the experiment with a higher concentration, but this peak is too short to be dimers' junction. In contrast, *l*-Th exhibit monomer's conductance and dimer's conductance during measurement, and the junction formation probability for dimer increase under intensified biases. The differences of conducting behaviors between linear *l*-Th and foldamers imply that intermolecular interaction is not detectable in foldamers.

Fig. R5 The upper panels are crystals packing with labelled intermolecular distance and torsion angles for *f*-Fu, *f*-Th and *l*-Th. The bottom panels are conductance measurement for *f*-Fu and *f*-Th with the concentration of 1mM and conductance measurement for *l*-Th with the concentration of 0.2 mM in THF: TMB (1:4, v/v).

8. There are some errors that need to be corrected, including but not limited to:

- 1) Statistical conductance data of *f*-Ph under different applied biases are not presented in Fig. 2j as described in the legend.
- 2) According to the insert in Fig.2f, the observed junction length of *f*-Th is 1.28 nm after adding 0.5 nm snap-back

distance, instead of 1.25 nm. Please check this value.

3) At the end of Page 4, “But the minor LC states of f-Fu own higher probabilities around 40% than the minor HC states of f-Th around 30%...”, are the minor states of f-Fu and f-Th HC and LC, respectively?

4) The legend of Fig. 4a (bottom panel) is missing.

5) In Page 11, the figure number should be Fig. 6, instead of Fig. 5.

6) Fig. 7a-d are not “2D conductance-displacement histograms”, and the corresponding color bars should be provided in these figures.

Answer: We thank reviewer’s detailed suggestions. The re-submitted manuscript has been thoroughly checked and revised.

1) Statistic conductance data of f-Ph are complemented in Fig 2f.

2) The observed junction length of f-Th is 1.28 nm after adding 0.5 nm snap-back distance. The mistaken value has been revised.

3) The minor states of f-Fu and f-Th induced by electric field are their corresponding HC and LC states, respectively, as f-Fu exhibits “turn-on” while f-Th exhibits “turn-off” features.

4) The missing legend is complemented.

5) The mistake in Page 11 has been revised.

6) Fig 7a-d are “2D conductance-time histograms”. The corresponding color bars are provided in the revised manuscript.

REVIEWERS' COMMENTS

Reviewer #1 (Remarks to the Author):

Below is the review response for answer to the 1st reviewer comments.

1. The expression concerns on "So" has been revised by the authors.
2. The typo has been revised by the authors.
3. The experiment setup and electrolyte selection have been discussed and added in the supporting information according to reviewer's suggestion.
4. This comment is well addressed by the authors.
5. The authors clearly explain to the reviewer's question.
6. Scale bar is added for figure 6.
7. F-Fu and f-Th is marked in Figure 1a.
8. Relevant explanation about the impurity is suggested to added in the main text. This point needs further revision.
9. The authors have delivered required discussions on the potential of solid-state devices based on the proposed foldamers, which will inspire other researchers for further extended work.

The authors have also replied detailed to other reviewers' comments and technique concerns.

In this manuscript, the authors have done detailed characterizations and scientific discussions on the design of foldamers as potential volatile memory elements. This is a well-written and prepared manuscript, which is qualified to be published in Nature Communications. After minor revision on question 8, this manuscript would be acceptable.

Reviewer #2 (Remarks to the Author):

All the comments have been fully addressed by authors. Therefore, the reviewer suggests the manuscript can be accepted by Nat. Commun.

Reviewer #3 (Remarks to the Author):

Zhao et al have comprehensively addressed my concerns, and I fully support publication now.

Response to Reviewer #1:

8. Relevant explanation about the impurity is suggested to added in the main text. This point needs further revision.

Answer: We thank the reviewer's good suggestion. The possibility of impurity-induced conductance switching is firstly ruled out before the discussion of other possibilities. Relevant explanation about the impurity is complemented in the subsection of "Electro-responsive *in-situ* conductance switching" of Results section.